

# The large mammal fossil fauna of the Cradle of Humankind, South Africa: a review

Megan Malherbe[1,2], Robyn Pickering[2,3], Deano Stynder[4] and Martin Haeusler[1]

[1] Institute of Evolutionary Medicine, University of Zürich, Zürich, Switzerland
[2] Human Evolution Research Institute, University of Cape Town, Cape Town, South Africa
[3] Department of Geological Sciences, University of Cape Town, Cape Town, South Africa
[4] Department of Archaeology, University of Cape Town, Cape Town, South Africa

## ABSTRACT

South Africa's Cradle of Humankind UNESCO World Heritage Site has remained the single richest source of hominin fossils for over ninety years. While its hominin specimens have been the subject of extensive research, the same is not true for its abundant faunal assemblages, despite their value in Plio-Pleistocene palaeoenvironmental reconstructions. Moreover, precise ages and depositional histories have been historically difficult to assess, though advancements in both relative and absolute dating techniques are changing this. This review explores the history of non-hominin large mammal faunal reporting, palaeoenvironmental reconstructions based on these fauna, and dating histories (with a focus on biochronology) at the following eight fossil-bearing sites of the Cradle that have been radiometrically dated with uranium-lead: Bolt's Farm, Cooper's Cave, Drimolen, Haasgat, Hoogland, Malapa, Sterkfontein and Swartkrans. Continued efforts to provide more precise and direct ages for sites using a variety of methods indicate that the bulk of Cradle deposits date to between 3 and 1.4 Ma. We find that, across almost all eight sites, there is little discussion or debate surrounding faunal reports, with some sites described by a single publication. Many of the reports are decades old with little review or reanalysis in the years following, emphasising the need for reviews such as this one. Our analysis of the data indicates that faunal-based paleoenvironmental reconstructions across sites commonly show a trend of wooded landscapes giving way to grasslands. We find that these reconstructions are primarily based on faunal abundance data, despite the availability of many other informative analytical techniques. The findings of this review highlight a need for more extensive and robust faunal reporting, as this will aid in understanding the context of these Cradle sites.

# INTRODUCTION

South Africa's Cradle of Humankind World Heritage Site (known locally as the Cradle), located in the northeast of the Gauteng province (Fig. 1), is home to some of the most important sites relevant to human evolution. Hominin fossils represented are attributed

Corresponding author
Megan Malherbe,
megan.g.malherbe@gmail.com

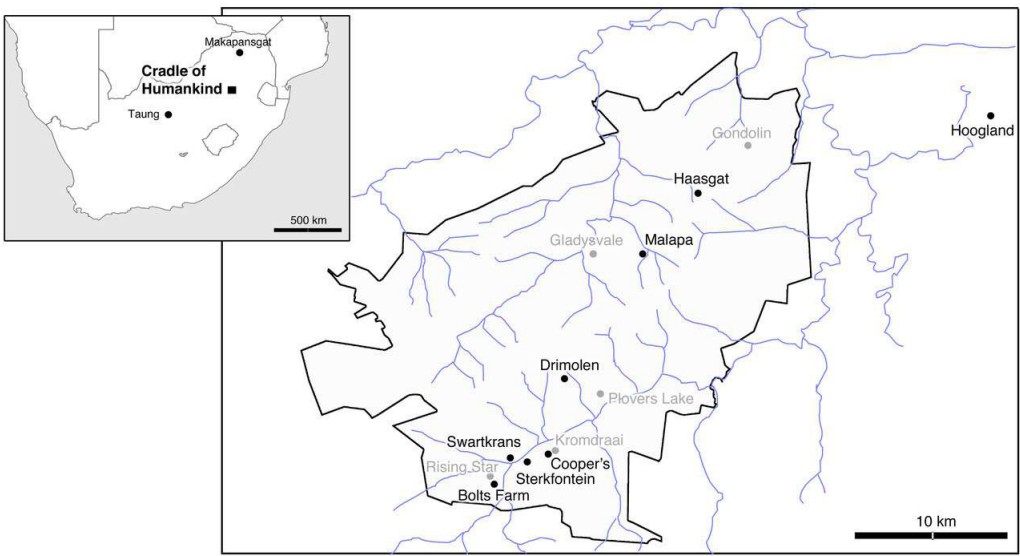

**Figure 1** Map of South Africa's Cradle of Humankind, with the eight key fossil sites included in this review shown in bold.

to at least six species—namely *Australopithecus africanus*, *A. prometheus*, *A. sediba*, *Paranthropus robustus*, *Homo erectus*, and *H. naledi*, and have been the subject of numerous studies ever since their discovery (*e.g., Berger et al., 2010*; *Brain et al., 1988*; *Clarke, 2013*; *Clarke, 2008*; *Clarke & Kuman, 2019*; *Lee-Thorp, Van der Merwe & Brain, 1994*; *Martin et al., 2021*; *Partridge et al., 1999*; *Pickering et al., 2012*; *Rak et al., 2021*; *Zanolli et al., 2022*).

In contrast to the hominin remains recovered at many Cradle sites, less attention has been paid to the faunal assemblages. A faunal report, which refers to a study that details faunal remains found at a site, is often focused on a single taxon or group, though this can vary across studies. A faunal report differs from a list or catalogue in that the latter is solely an inventory of specimens, whereas the former includes descriptions of species, distribution within the site with regards to other taxa, ecological and biochronological interpretations, and any other relevant information that can aid overall understanding of the site. Many of the faunal reports from the Cradle have not been explored further or expanded upon since their initial publication (*e.g., De Ruiter, 2003* for Swartkrans), or the reports are limited in terms of the taxa that are investigated, which can in turn limit biochronological deductions. There is an increased risk that specimens would go missing if faunal reporting is delayed for years after the site and the assemblages are first mentioned. This has been evidenced with two carnivore fossils and an additional 21 initially recorded specimens from Haasgat, 19 of which were bovids (*Adams, 2012*; *Keyser, 1991*). Moreover, faunal reports are often later found to be incomplete or contradictory (*Adams, 2012*; *Adams et al., 2016*; *De Ruiter et al., 2009*; *Edwards et al., 2019*), highlighting a lack of cohesive reporting at times. This, by extension, highlights the potential for still-unrecognised incongruities in faunal assemblages and reports that have gone without analysis or re-examination since their first mentions. Lastly, issues surrounding historical faunal reports in general are

**Table 1 Commonly found bovid tribes at Cradle sites and their associated trophic and habitat categories.** Trophic assignments follow *Sponheimer, Reed & Lee-Thorp (1999)* and *Steininger (2011)*. Habitat categories follow *Plummer et al. (2015)*.

| Bovid tribe | Feeding category | Habitat category |
|---|---|---|
| Alcelaphini | Mixed feeder-grass | Light cover |
| Tragelaphini | Mixed feeder-browse | Heavy cover |
| Bovini | Graze | Light cover |
| Hippotragini | Mixed feeder-grass | Open |
| Reduncini | Fresh grass | Light cover |
| Peleini | Mixed feeder-browse | Light cover |
| Antilopini | Mixed feeder-grass | Open |
| Aepycerotini | Mixed feeder-grass | Light cover |
| Neotragini | Mixed feeder-browse | Light cover |
| Ovibovini | Mixed feeder-grass | Light cover |
| Cephalophini | Browse | Forest |

consistently confounded by the fact that species attributions, stratigraphic associations or element identification can be revised after their initial publication (*Allmon & Yacobucci, 2016*; *Badenhorst & Steininger, 2019*; *Bennett, 1980*; *Clarke, 2019*; *Cooke, 1950*; *Demeré, 1986*). This further underscores the importance of reassessing and updating faunal reports when possible, as well as reporting on all taxa that are widely represented at a site.

Having complete and extensive faunal reports of Plio-Pleistocene sites is valuable for several reasons. They aid in palaeoenvironmental reconstruction and can inform on aspects such as vegetation types and climatic conditions. Thorough faunal reporting is also crucial for biochronology, an integral technique in reconstructing the timeline of hominin history. As certain species have well-established temporal ranges, studying a wider variety of species enhances the accuracy of determining the faunal age of a site. Patterns of evolution are also better understood with the aid of accurate faunal reports, as they contribute to our understanding of changes in biodiversity over time. Faunal reporting can also yield inferences about early hominin behaviour, as specific archaeozoological and taphonomic studies based on comprehensive faunal reports can help to identify agents of bone accumulation.

## Fauna as palaeoenvironmental and chronological indicators

Fossil fauna are frequently used to estimate palaeoenvironmental conditions at Cradle sites. Bovids are particularly useful for this (*Bobe & Eck, 2001*; *Gentry, 1970*; *Kingston, 2007*; *Plummer, Bishop & Hertel, 2008*; *Shipman & Harris, 1988*; *Vrba, 1980*) as they occupy a wide range of habitats on the continent, including semi-deserts, shrublands, open grasslands, woodlands and forests, and species are both habitat and diet-specific (*Sponheimer, Reed & Lee-Thorp, 1999*; *Vrba, 1975*; *Vrba, 1980*) (Table 1)—although some exhibit flexibility in this respect (*Blondel et al., 2022*; *Codron, Hofmann & Clauss, 2019*; *Sewell et al., 2019*).

Research has focused on fossil herbivore diets and habitat preferences, as well as community composition as palaeoenvironmental proxies, determining the types of

vegetation consumed and what habitats animals occupied. Establishing the dietary proclivities of extinct herbivores can be achieved in various ways, namely isotopic studies of $\delta^{13}$C in tooth enamel, which present the ratio of $C_4$ *vs.* $C_3$ plant consumption in a given individual (*Lee-Thorp, Sponheimer & Luyt, 2007*; *Sponheimer & Lee-Thorp, 2003*; *Sponheimer, Reed & Lee-Thorp, 2001*; *Uno et al., 2011*). Dental mesowear is used to evaluate feeding behaviours through gross occlusal wear patterns of molar teeth (*Blondel et al., 2010*; *Dumouchel & Bobe, 2020*; *Kaiser & Fortelius, 2003*; *Sewell et al., 2019*; *Stynder, 2011*). Dental microwear examines occlusal tooth surfaces at high magnification to determine types of vegetation consumed weeks before death (*Fortelius & Solounias, 2000*; *Merceron et al., 2005*; *Merceron & Ungar, 2005*; *Schubert et al., 2006*; *Ungar, Scott & Steininger, 2016*). Reconstructing ecological circumstances at hominin sites also frequently makes use of ecological morphology (or ecomorphology), a technique which examines how animals are functionally adapted to their surroundings (*Barr, 2014*; *Bishop et al., 2011*; *Elton, 2001*; *Forrest, Plummer & Raaum, 2018*; *Kovarovic et al., 2021*; *Plummer & Bishop, 1994*; *Sambo, 2020*). The method typically focuses on postcranial elements that are challenging to identify to species level, often resulting in these data being absent from faunal reports, which prioritize taxonomically identifiable elements. The overall structure of animal communities, represented by diversity or dominance of species, is also utilised as a vegetative signal (*Dodd & Stanton, 1990*; *Reed, Spencer & Rector, 2013*). This most typically entails analysing the number of individuals of a taxon relative to the total assemblage, and presence/absence of taxa (*Alemseged, 2003*; *Greenacre & Vrba, 1984*; *Hanon et al., 2022b*; *Vrba, 1975*; *Vrba, 1980*). It is important to note, however, given the time- and climate-averaged nature of Cradle deposits (*Behrensmeyer, Kidwell & Gastaldo, 2000*; *Behrensmeyer & Reed, 2013*; *Hopley & Maslin, 2010*), that the relationship between specific indicator taxa and certain habitat conditions is not necessarily as clear as widely assumed (*Negash & Barr, 2023*; *Sokolowski et al., 2023*).

Besides palaeomagnetic analyses (*Partridge et al., 2003*; *Partridge et al., 1999*), the dating of Cradle sites has primarily been done *via* faunal comparisons (*e.g.*, *Berger et al., 2003*; *Berger, Lacruz & De Ruiter, 2002*; *Delson, 1984*; *Delson, 1988*; *Dirks et al., 2010*; *Keyser, 1991*; *McKee, 1991*; *Pickford & Gommery, 2020*; *Rovinsky et al., 2015*; *Steininger, Berger & Kuhn, 2008*; *Turner, 1997*; *Vrba, 1975*; *Vrba, 1995*), often with well-dated eastern African sites (*Brown et al., 1985*; *Deino, 2011*; *Deino & Hill, 2002*; *Deino et al., 2010*; *Deino et al., 2002*; *Gathogo & Brown, 2006*; *McDougall & Brown, 2008*; *Walter et al., 1991*; *WoldeGabriel et al., 1992*) since suitable material for absolute dating methods like argon-argon or potassium-argon is absent in South Africa. This biochronological method does come with drawbacks, mainly due to the difficulties in precisely determining the timing of extinction/speciation events, and the potential for mixing or displacement of remains between deposits at the Cradle. Biochronology nevertheless also offers advantages over other approaches. Unlike speleothems, *in situ* fauna are found in direct association with hominins and occupy the same overall infill. Therefore, even if a deposit is time-averaged or mixed, the overall assemblage would have accumulated *via* comparable processes. Moreover, although other methods are sometimes able to provide narrower age estimates than fauna, they do so by requiring more modelling and assumptions, which can be challenging to

estimate. Faunal comparisons have been utilised for many years, particularly at Sterkfontein and Swartkrans (*De Ruiter, 2003*; *Vrba, 1975*; *Watson, 1993*). While comparisons to eastern African assemblages are useful and indeed helped to develop the initial chronology for South Africa's cave sites (*De Ruiter, 2003*; *Delson, 1988*), the two regions are up to 4,000 km apart and are subject to vastly different climatic and environmental conditions (*King & Bailey, 2006*; *O'Brien & Peters, 1999*; *Partridge, Wood & De Menocal, 1995*). Additionally, South Africa has been described as somewhat of a refugium where animal taxa tended to linger, while their eastern African counterparts had already gone extinct (*Arctander, Johansen & Coutellec-Vreto, 1999*; *Bailey, King & Manighetti, 2000*; *Reynolds, 2007a*). Thus, different evolutionary scenarios may have existed across Africa during the Plio-Pleistocene, with the South specifically being more stable. Strictly evaluating such a hypothesis, however, is presently challenging due to insufficient faunal reporting, and at times dating, across South African sites.

Our understanding of Plio-Pleistocene South Africa is now being improved by combining biochronological and geochronological techniques. Uranium-lead (U-Pb) dating of speleothems (cave carbonates) has, most recently, been used to illustrate that flowstones of the same age occur across various Cradle cave deposits (*Pickering et al., 2019*). These flowstones, present at all Cradle sites, are thus able to provide a temporal framework for hominin occupation, similar to the volcanic tuffs in eastern Africa (*Pickering & Herries, 2022*).

In light of the U-Pb chronologies for Bolt's Farm, Cooper's Cave, Drimolen, Haasgat, Hoogland, Malapa, Sterkfontein and Swartkrans (*Pickering et al., 2019*), the following review explores the published faunal reports for these eight localities, highlighting a shortage in faunal reporting across many of them. This particular subset of Cradle sites allows us to evaluate the biochronological estimates in light of the absolute ages. Particular attention is paid, where possible, to the representation and reporting of bovid taxa, due to their well-established abundance at Plio-Pleistocene hominin sites (Fig. 2) and their value in palaeoenvironmental reconstructions (*Barr, 2014*; *Behrensmeyer et al., 1997*; *Bibi & Kiessling, 2015*; *Bishop et al., 2011*; *Bobe & Eck, 2001*; *Lee-Thorp, Sponheimer & Luyt, 2007*; *Vrba, 1993*). We also explore the history of dating in the Cradle, both relative and absolute, and illustrate how assigned ages have been reformed with the introduction of new or different methodologies. We will discuss the palaeoenvironmental reconstructions based on large mammal fauna that have been proposed for Cradle sites, while highlighting the need for more comprehensive reconstruction techniques at a wider range of Cradle sites. We will also demonstrate how the faunal accounts and associated biochronological dating have influenced age assignments of these sites, and how a multi-chronological approach has provided the more robust chronological framework that we now recognise.

## SURVEY METHODOLOGY

U-Pb dating of the Cradle caves presents hypotheses relating to regional climate and vegetation cycles (*Pickering et al., 2019*), therefore here we focus on the sites dated by U-Pb (listed alphabetically): Bolt's Farm, Cooper's Cave, Drimolen, Haasgat, Hoogland, Malapa,

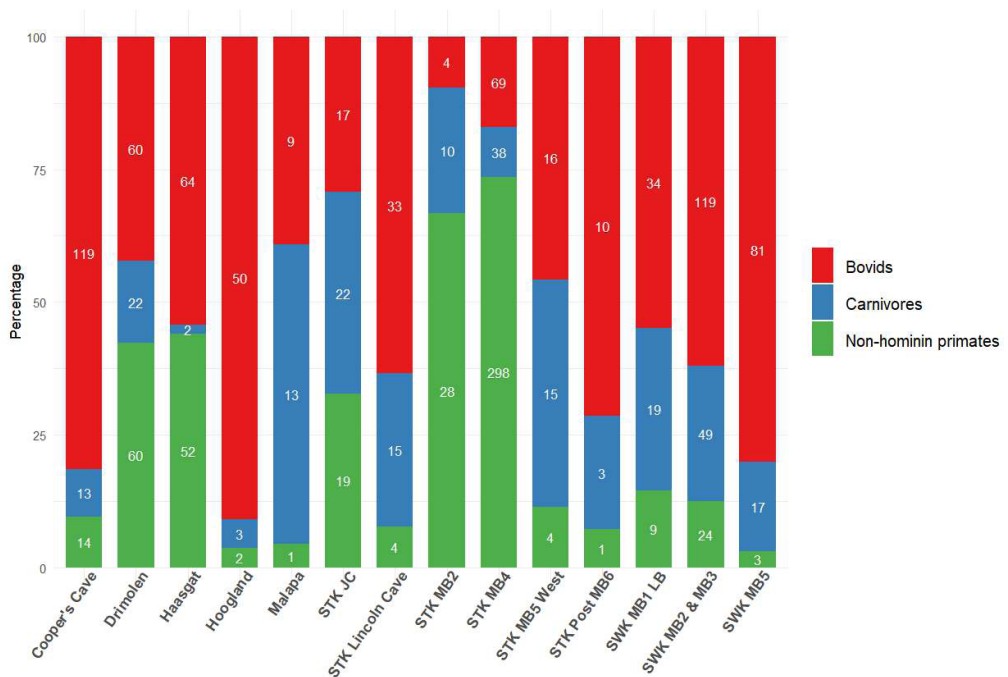

**Figure 2  Bovid, carnivore and non-hominin primate abundance within different Cradle deposits.**
Minimum number of individuals (MNI) shown inside bars. Numbers come from most recent
publications, see Table S1. Bolt's Farm not included due to insufficient records for the three taxa across pit
localities. MB, Member; LB, Lower Bank; JC, Jakovec Cavern; LC, Lincoln Cave.

Sterkfontein, and Swartkrans. The *Pickering et al. (2019)* hypothesis states that the faunal
records of the Cradle are inherently biased towards the presence of dry-adapted fauna,
as the caves were open for sediment/fossil accumulation only during drier times. This
has implications both for how the faunal communities of these eight sites are assessed,
as well as interpretations of extinction or speciation events. Here we aim to test this
hypothesis, drawing on unpublished PhD theses, book chapters, and peer-reviewed journal
articles. Relevant publications were identified using variations of the keywords 'fossil
fauna', 'Cradle of Humankind South Africa', 'vertebrate mammals', 'palaeoenvironmental
reconstruction', 'absolute dating', 'faunal biochronology', and the relevant site names in
Google Scholar, ResearchGate, and South African university repositories (for the theses).
Master's theses were not included unless they were specifically centred around the faunal
communities of one or more of the sites. As large mammals typically represent the most
abundant fauna at Cradle sites, this is the core focus. Thus, herpestids, amphibians, and
other non-mammalian fossils are not discussed. Studies on micromammals and birds are
very briefly addressed but are mainly excluded from this review.

# SITE HISTORIES

## Bolt's farm

### Faunal reporting

*Broom (1937)* conducted the initial research at Bolt's Farm and continued to sample the area until 1948. Since that time, it has become evident that the locality is made up of over 30 separately eroded palaeocave remnants, with different 'pits' revealed from lime mining and often deemed to differ in age and faunal composition (*Edwards et al., 2019*; *Edwards et al., 2020*; *Monson, Brasil & Hlusko, 2015*). The pits and caves that make up Bolt's Farm today occur on three properties, and deposits are thought to have formed as part of the same cave system but different times (*Gommery et al., 2012*), or as entirely separate caves. *Edwards et al. (2019)* provide spatial data and survey control points to reduce confusion surrounding the site, as various deposits at Bolt's Farm have historically been given different names across publications (*e.g.*, Bridge Cave, Elephant Cave and Pit 7 all refer to the same locality). Various type specimens were described upon the site's discovery, mostly of carnivores (*Broom, 1939*), though these have since been subsumed or are reported as no longer locatable (*Badenhorst et al., 2011*; *Edwards et al., 2019*). Moreover, the origin of these early discoveries remains ambiguous. With no recorded information on precise pit deposits, early reports are unclear and challenging to address (*Edwards et al., 2019*). Bolt's Farm was later excavated by the University of California African Expedition in 1947/8, which uncovered various small deposits of bone-bearing sediment (*Cooke, 1993*; *Thackeray et al., 2008*). The majority of the collected material lacked detailed descriptions, and the collection localities of specimens, whether they were *in situ* or from miner's dumps, were not specified. The exception to this was one mustelid and material described as originating from *Dinofelis barlowi* (*Cooke, 1991*). The Waypoint 160 deposit, first discussed by *Senegas & Avery (1998)*, yielded a set of fragmented specimens identified as *Parapapio* (*Gommery et al., 2008*), but as of yet, no hominins have been discovered in any of the Bolt's Farm pits. Over fifty years after the initial excavation, researchers began to examine and describe selected fauna and their associated environments, particularly in the case of cercopithecoids (*Elton, 2001*; *Freedman, 1957*; *Freedman, 1965*; *Gommery et al., 2008*; *Gommery et al., 2009*) and two *Antidorcas* specimens from Pit 3 (*Cooke, 1996*; *Reynolds, 2007b*). Various extinct species of suids were excavated from across five karst deposit localities at Bolt's Farm and reported on in some detail (*Pickford & Gommery, 2016*; *Pickford & Gommery, 2020*). Faunal lists composed of specimen numbers and taxonomic orders were provided for pits 1–8, 10, 14–16 and 23 (*Monson, Brasil & Hlusko, 2015*). *Badenhorst et al. (2011)* reported findings from an excavation conducted in 2003 at the Garage Ravine (or Pit 4) locality at Bolt's Farm, and to date, this appears to be the most comprehensive faunal reconstruction for part of the overall area. Fauna from X Cave (or Pit 11) is presented in similar detail (*Van Zyl, Badenhorst & Brink, 2016*), though this report solely focused on bovid specimens and did not include other families. For the most up to date faunal descriptions for each Bolt's Farm locality, see *Edwards et al. (2019)* Supplementary data Text S1 (https://opal.latrobe.edu.au/articles/dataset/Edwardsetal_BFSOM/7238126/1).

### Biochronology and absolute dates

Despite the small sample size of 10 specimens, the fauna from the Garage Ravine excavation was used to infer potential ages of the fossil bearing sediments, broadly estimated to be younger than 2 Ma but older than 10 ka (*Badenhorst et al., 2011*). The taxonomic status of the equids has been debated (*Thackeray, 2010*), and by extension the age of the area—large size was the primary basis for ascribing fossil teeth to *Equus capensis* rather than *Equus quagga*, though this has since proven a contentious method for distinguishing between the two species (*Lorenzen et al., 2010*; *Malherbe, 2019*). With regards to primates, *Theropithecus oswaldi leakeyi* suggested an age of 1.9–0.7 Ma for pit 10, *Papio angusticeps* and *Cercopithecoides coronatus* suggested 2.1–1.6 Ma for pit 6, and *Papio robinsoni* and *Cercopithecoides williamsi* suggested 2.6–2.0 Ma for pit 23 (*Frost et al., 2022*). In contrast, suids recovered from various Bolt's Farm localities (Pit 3, Pit 14, Milo A, Pit 7, Alcelaphine Cave, Brad Pit A and Pit 15) indicated ages between 3.7–1.8 Ma (*Pickford & Gommery, 2020*), with Milo A revealing a *Metridiochoerus andrewsi* specimen, alluding to a 3.04–2.58 Ma age (*Gommery et al., 2012*).

Bolt's Farm's X Cave was regarded as problematic as it consists of deposits that may be of various ages (*Van Zyl, Badenhorst & Brink, 2016*). *Connochaetes* cf. *gnou* was identified at the site, found alongside *Antidorcas bondi* and *Aepyceros melampus*, leading to the assumption that these species were contemporaneous. The first appearance of *C. gnou* in the palaeontological record is around 1.17 Ma at Cornelia-Uitzoek in the Free State province, where it was also recovered with *An. bondi* and *Ae. helmoedi* (*Brink et al., 2012*). X Cave was therefore noted as having had a similar environment to Cornelia-Uitzoek, though it was presumed to be younger (*Van Zyl, Badenhorst & Brink, 2016*), with no justification provided for this assumption. *Edwards et al. (2019)* noted that the faunal data were insufficient from pits 2, 8, 15, 17, Jackal Cave and Brad Pit A and B, hence establishing an age bracket for these deposits was not feasible. For pits 1, 3, 6 and 25, a minimum age of 0.78 Ma was suggested due to presence of *Antidorcas recki*, a bovid that disappeared from South African deposits after the formation of the Western Cape open-air locality of Elandsfontein (*Klein et al., 2007*; *Klein & Cruz-Uribe, 1991*). However, presence of *Cercopithecoides coronatus* in Pit 6 would suggest a minimum age of ∼1.5 Ma (*Delson, 1984*; *Frost et al., 2022*). The three-toed equid *Eurygnathohippus* was found in Pit 16, which suggested that it had likely been deposited prior to 0.99 Ma. There is no distinct fauna that is able to constrain the minimum depositional age for pits 4, 5, and New Cave. *Parapapio* specimens from Waypoint 160 indicated an age between 4.5–4.0 Ma (*Gommery et al., 2008*), though these specimens are not diagnostic below genus level. A rodent species (*Euryotomys bolti*) initially suggested an age of between 5–4 Ma (*Senegas & Avery, 1998*), though this has since been challenged as its provenience was never known and the pit itself was subsequently radiometrically dated (*Edwards et al., 2023*).

Thus, U-Pb dating of Waypoint 160 presented an overall younger age of 2.27–1.7 Ma (*Edwards et al., 2023*). A combination of palaeomagnetism and U-Pb data suggested that the Aves Complex (comprised of pits 5, 8 and 14) dates to between 3.03–2.61 Ma (*Edwards et al., 2020*). There are currently no other geochronological data available for the other pits at Bolt's Farm, so fauna has remained the primary chronological indicator.

## Cooper's cave
### Faunal reporting

The Cooper's Cave site is subdivided into three distinct localities, designated Cooper's A, B and D, and it was noted that there were no significant faunal differences between them (*Berger et al., 2003*) and thus they most likely represent one unit of time. However, no direct faunal evidence was provided to support this hypothesis. Moreover, fossils initially discovered at Cooper's A and B during excavations in the 1950s were later found to not definitively originate from those deposits (*De Ruiter et al., 2009*). After the discovery of a since-lost hominin molar (*De Ruiter et al., 2009*) attributed initially to *Homo* and then to *A. africanus* in one of these deposits (*Broom & Schepers, 1946*; *Shaw, 1940*), subsequent investigations (*Brain, 1958*) in both localities gradually faded, until renewed explorations began again in the 1990s. Cooper's Cave has since yielded various early *Homo* and *P. robustus* specimens (*Berger, Pickford & Thackeray, 1995*; *De Ruiter et al., 2009*; *Shaw, 1940*; *Steininger, Berger & Kuhn, 2008*). *Freedman (1957)* described the primates from Cooper's A in detail, and *Papio* material from this locality was later reviewed (*Gilbert et al., 2018*). However, the majority of fossil fauna has been recovered from Cooper's D, and, as a result, almost all publications refer to this sequence alone. Bovids dominate the site numerically, and specimens from every major tribe, Alcelaphini, Antilopini, Bovini, Hippotragini, Neotragini, Ovibovini, Peleini, Reduncini and Tragelaphini were found (*Hanon et al., 2022a*; *Hanon et al., 2022b*; *Steininger, 2011*) (Table S2). Although an exhaustive study of the bovid assemblage was recently published (*Hanon et al., 2022b*), studies have mostly focused on the carnivores, as is common at Cradle sites (Fig. 3). Reanalysis of the faunal assemblage by *De Ruiter et al. (2009)* noted that in fact over 50,000 specimens had been catalogued and not just 9,000 as initially reported (*Berger et al., 2003*), though it was still agreed that the sequences appear homogenous. The reason for this was that the material analysed came from decalcified sediments—they were thus considered as one entity "pending more detailed analysis" (*De Ruiter et al., 2009*: 504). Since this time, the primates (*DeSilva, Steininger & Patel, 2013*; *Folinsbee & Reisz, 2013*; *Val, Taru & Steininger, 2014*), Hyaenidae (*Kuhn, Werdelin & Steininger, 2017*), Felidae (*Hartstone-Rose et al., 2007*; *O'Regan & Steininger, 2017*), Canidae (*Hartstone-Rose et al., 2010*), Equidae (*Badenhorst & Steininger, 2019*), Bovidae (*Hanon et al., 2022b*) and other taxa (*Cohen, O'Regan & Steininger, 2019*; *O'Regan, Cohen & Steininger, 2013*) have been reported on in greater detail and utilised in palaeoenvironmental reconstructions of the site.

### Biochronology and absolute dates

A palaeontological age of 1.9–1.6 Ma was assigned to the overall deposit, based on broad correlations with fauna from Swartkrans and Kromdraai A (*Steininger, Berger & Kuhn, 2008*); sites that were themselves dated biochronologically or *via* palaeomagnetism (*De Ruiter, 2003*; *Herries, Curnoe & Adams, 2009*; *Kuman, Field & Thackeray, 1997*; *McKee, Thackeray & Berger, 1995*; *Vrba, 1995*). Cercopithecids from Cooper's A suggested an age of ∼2.0–1.5 Ma (*Delson, 1984*), and primate biochronology recently suggested a range of 1.9–1.6 Ma for Cooper's D (*Frost et al., 2022*). U-Pb dating of the flowstones at Cooper's D provided a narrower age bracket of 1.5–1.4 Ma for the majority of the assemblage

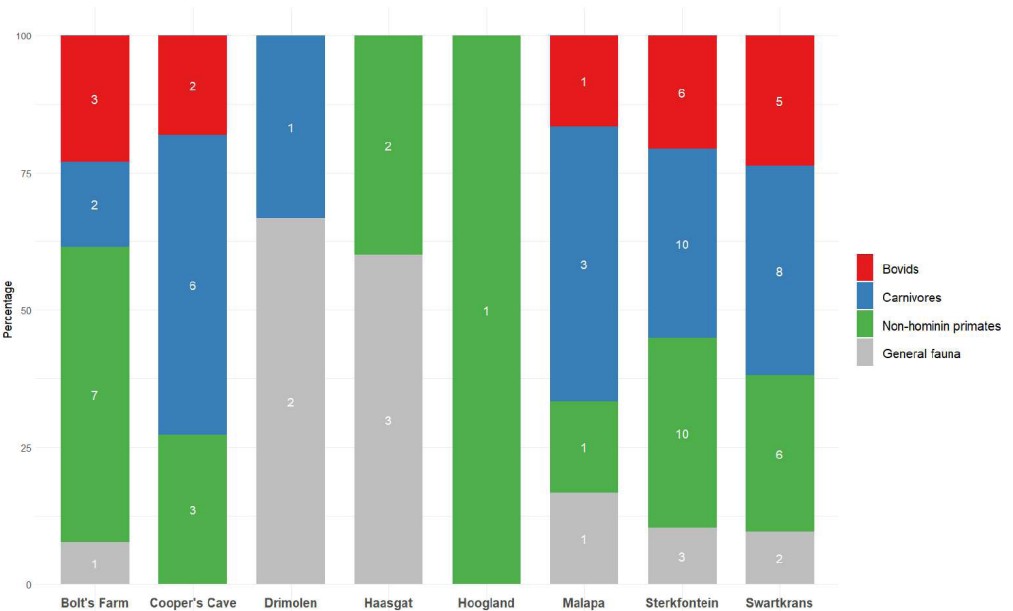

**Figure 3  Publications dedicated to bovids, carnivores or non-hominin primates of eight Cradle sites.**
Number of publications shown inside bars, see Table S3. Publications were not included if they were based on multiple sites (more than 3). 'General fauna' refers to the overall mammalian assemblage at a site.

(*De Ruiter et al., 2009*). Many of the fossils were recovered from sediments above the dated flowstone, thus they are likely younger than 1.4 Ma, such as the isolated hominin molars (*De Ruiter et al., 2009*). *Pickering et al. (2019)* later recalculated this 1.4 Ma age to <1.3 Ma.

## Drimolen
### *Faunal reporting*
The Drimolen system, first excavated in the 1990s, is home to the most complete cranium of *P. robustus* to date, DNH 7 (*Keyser et al., 2000*; *Rak et al., 2021*), and the *H. erectus* cranium DNH 134 (*Herries et al., 2020*), confirms the coexistence of these two genera (*Keyser et al., 2000*; *Moggi-Cecchi et al., 2010*). Two distinct areas are recognised at Drimolen, the Main Quarry (DMQ) and the Drimolen Makondo (DMK), the latter of which is an isolated solution-tube of decalcified matrix which was known but not sampled until 2013. The deposits were revealed to have different faunal assemblages, leading to suggestions that the areas likely differ in age (*Rovinsky et al., 2015*). Despite the site's year of discovery is recorded as 1992 (*Keyser et al., 2000*), the first description and analysis of non-primate macromammalian fauna, collected from DMQ, was published over twenty years later (*Adams et al., 2016*). When DMK was excavated, it also yielded a diverse macromammalian sample (*Rovinsky et al., 2015*) which included primates (Table S4). However, palaeoenvironmentally and chronologically informative taxa like suids or equids were largely absent. A provisional carnivore species list and an analysis of carnivores recovered prior to 2008 from DMQ were reported in *O'Regan & Menter (2009)*. However, later examination of all macromammals revealed the carnivore assemblage to be more

diverse than previously established (*Adams et al., 2016*), with four additionally recognised species included.

### Biochronology and absolute dates

Few of the recovered faunal specimens from Drimolen are time-sensitive and able to provide chronological insight, particularly due to the lack of suid and equid remains. Despite this, DMQ was still assigned a faunal age estimate of 2.0–1.5 Ma based on its correlation to Swartkrans Member 1 and Cooper's Cave (*Rovinsky et al., 2015*), the co-occurrence of *Homo* and *Paranthropus* at the site (*Moggi-Cecchi et al., 2010*), and the overall mammalian assemblage—though without further explanation regarding this (*Keyser et al., 2000*; *O'Regan & Menter, 2009*). More recently discovered carnivores from DMQ have had implications for biochronology—*Chasmaporthetes silberbergi* and *Dinofelis* cf. *barlowi* implied an age of pre-2.0–1.8 Ma, and *Dinofelis* aff. *piveteaui* suggested a maximum depositional age of 1.6 Ma (*Adams et al., 2016*), all based on their presence in eastern Africa. Other DMQ fauna are typically part of long-surviving Plio-Pleistocene lineages, or have poorly-secured first appearance dates (FADs) and last appearance dates (LADs) (*O'Regan & Menter, 2009*) which limit biochronological interpretations. The non-hominin primates found at the site, attributed to *Papio hamadryas robinsoni* and *Cercopithecoides williamsi* (*Adams et al., 2016*), have broad but informative age constraints of ∼2.3–1.6 Ma and ∼2.7–1.6 Ma respectively (*Frost et al., 2022*). DMK, considered to be older than DMQ (Fig. 4) lacks several taxonomic groups that are well represented at DMQ, most notably hominins and *Papio*. There is an overall lack of identifiable and biochronologically sensitive species at DMK, though a *Metridiochoerus* premolar indicate a potentially older age for this deposit than the Main Quarry (*Rovinsky et al., 2015*). Although bovid taxa comprise 78% of the identifiable DMK faunal assemblage, they were deemed to be of little value in determining a biochronological age due to the lack of time-specific species (*Rovinsky et al., 2015*). Relying on palaeomagnetic, electron spin resonance (ESR) and U-Pb data has thus been crucial in refining the depositional history of DMK (*Murszewski, Boschian & Herries, 2020*).

The first direct dates for Drimolen came from ESR analyses of bovid teeth, which provided an age of 1.7 Ma for DMQ and 2.7 Ma for DMK, consistent with the faunal dating (*Herries et al., 2019*; *Herries et al., 2018*; *Herries et al., 2020*). U-Pb analysis provided similar ages of 2.62–2.28 Ma for DMK and 2.0–1.82 Ma for DMQ (*Pickering et al., 2019*). A combination of methods constrained the age of DMQ further, as a ∼1.95 million-year-old magnetic field reversal was identified within the sediments, and specified a narrow depositional age of 2.04–1.95 Ma (*Herries et al., 2020*; *Martin et al., 2021*).

## Haasgat
### Faunal reporting

In contrast to most other Cradle sites, the Haasgat palaeocave system is situated in a region with significant modern topographic relief. As a result of lime mining in the early 20th century and the subsequent collapsing of calcified sediment (*Herries et al., 2014*), no accurate *in situ* context exists for the majority of recovered specimens from Haasgat. The first hominin specimen from the site, HGT 500, consists of a partial maxillary molar

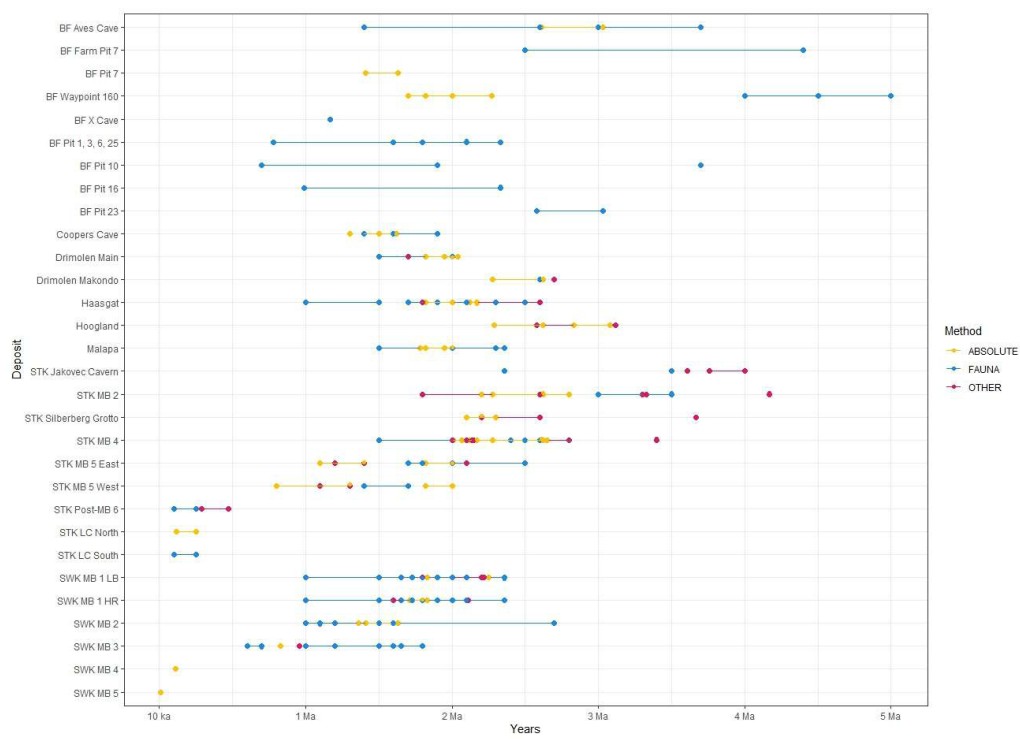

**Figure 4  All proposed dates for the various deposits at each Cradle site.** Refer to Table 2 for associated publications for each date. 'Other' refers to non-biochronological relative methods (*i.e.,* palaeomagnetism, ESR, cosmogenic nuclides). Date for Hoogland by *Hopley et al. (2019)* not shown here. BF, Bolt's Farm; STK, Sterkfontein; MB, Member; SWK, Swartkrans; LB, Lower Bank; HR, Hanging Remnant.

postulated to be within the *Australopithecus*-early *Homo* hypodigm (*Leece et al., 2016*). *Ex situ* fossils were first recovered in 1988 from a talus slope containing fossils and sediment and generated by the mining, and the first preliminary faunal report followed shortly after (*Keyser, 1991*). This initial report, which noted that *Cercopithecoides williamsi* was the most abundant of all taxa in the deposit, was otherwise fairly limited and focused mainly on primate craniodental specimens. Subsequent publications also focused solely on primates, specifically *Papio* (*McKee & Keyser, 1994*). A later taphonomic analysis of the site concluded that the overall high primate count could be the result of excavator bias (*Adams & Rovinsky, 2018*). Similarly, primates have been notably utilised in palaeoenvironmental considerations (*Adams & Rovinsky, 2018*; *Keyser, 1991*). In terms of other fauna, two carnivores that had initially been mentioned were never formally studied, and have since gone missing from the collection (*Adams & Rovinsky, 2018*). It was also originally noted that many large bovid bones were recovered from the dump (*Keyser, 1991*), yet no associated teeth and thus no identification was possible. Where teeth were found, fragmentation and damage were so bad that these too did not aid in identification to species or even tribal level—save for one alcelaphin molar (*Keyser, 1991*). Additional cercopithecoids were presented by *McKee, Von Mayer & Kuykendall (2011)*, who found that the material was distinct from all

cercopithecoid species found in eastern Africa and so represented a new species. In 2010, the cave system was reopened and the original assemblage was evaluated for the first time since the 1990s (*Adams, 2012*). *Adams (2012)* presented a revised faunal list from the site, closely examining the non-primate mammals for the first time and adjusting specimen counts and identifiable species. The review of the existing collection evidenced various unrecorded specimens, of which 83% were bovids. Many of the originally described specimens could not be located at all, nearly twice the number of specimens were newly noted, and many taxonomic classifications required revising. These substantial changes ultimately altered the overall faunal list (Table S5) and, consequently, the biochronology of the site (*Adams, 2012*).

### Biochronology and absolute dates

Initial reports on Haasgat noted that an insufficient number of fossils was identified and analysed, and therefore a potential age estimate for the deposit was not possible (*Keyser, 1991*). A preliminary age of ∼2.5 Ma was suggested based solely on the presence of *Parapapio* and *Chasmaporthetes* (*Keyser, 1991*). *Plug & Keyser (1994)* later argued that the overall species composition suggested an age no older than 1.5 Ma, a million years younger (Fig. 4). The reasoning for this was that while the bovids suggested an age of roughly 1 Ma, cercopithecoids and *Papio angusticeps* suggested one closer to 1.5 Ma. Two bovid species belonging to *Tragelaphus* were identified and said to be morphologically similar to extant relatives (*Plug & Keyser, 1994*). *Equus capensis, Procavia transvaalensis, Megalotragus priscus*, and *Antidorcas bondi* were noted as the only (recently) extinct species present at the site, and all are present in early archaeological deposits in southern Africa (*Brink, Holt & Horwitz, 2016*; *Brink et al., 1995*; *Ecker & Lee-Thorp, 2018*; *Kaiser & Franz-Odendaal, 2004*).

   In his review of the fauna, *Adams (2012)* argued that the most common bovid genus in the assemblage, which was previously not recognised, *Oreotragus*, differs morphologically from those at the nearby site of Gondolin, and thus a contemporaneous age of 1.8 Ma was discounted (*Adams, 2012*). *Adams (2012)* went on to suggest an age of between 2.3 and 1.9 Ma based on the presence of *Equus*, with other fauna such as *Connochaetes gnou* indicating a potentially younger, mid- to late-Pleistocene age. Due to the site's position as an *ex situ* assemblage with sampling and preparation techniques that have been viewed as biased towards primate and craniodental remains, the biochronology has been regarded as tentative (*Adams, 2012*; *Adams & Rovinsky, 2018*). Palaeomagnetic analysis of the oldest layers yielded a reversed polarity, with a normal polarity for the younger layers (*Herries et al., 2014*). With help from the age ranges suggested by the fauna, *Herries et al. (2014)* argued that the reversed polarity dates to 2.58–1.95 Ma, and normal polarity dates to 1.95–1.78 Ma. U-Pb dating of one flowstone from the middle of the sedimentary sequence at Haasgat provided an age of 1.686 ± 0.236 Ma (*Pickering et al., 2019*).

### Hoogland
### Faunal reporting

Hoogland is a relatively under-studied site (*Adams et al., 2010*), which has not yet produced any hominin specimens, though it has produced various habitat-specific bovid species as

**Table 2  Published dates for Cradle deposits using various methods.** Asterisk before the date indicates absolute methods were used. If a combination of relative and absolute methods was used, asterisk is shown. Dating efforts within deposits are ordered according to publication year. CN, cosmogenic nuclide dating; PM, palaeomagnetic analysis.

| Site | Published dates | Authors |
|---|---|---|
| **Bolt's Farm:** Aves Cave Complex | 3.7–1.4 Ma (carnivores, bovids, suids) | *Edwards et al. (2020)* |
| | 3.0–2.6 Ma (suids) | *Pickford & Gommery (2020)* |
| | *3.03–2.61 Ma (U-Pb & PM) | *Edwards et al. (2020)* |
| **Bolt's Farm:** Pit 7 | 4.4–2.5 Ma (fauna) | *Edwards et al. (2019)* |
| | *1.63–1.41 Ma (U-Pb) | *Pickering et al. (2019)* |
| **Bolt's Farm:** Waypoint 160 | 5–4 Ma (rodents) | *Senegas & Avery (1998)* |
| | 4.5–4.0 Ma (microfauna) | *Gommery et al. (2008)* |
| | <5 Ma (fauna) | *Edwards et al. (2019)* |
| | *2.0–1.82 Ma (U-Pb) | *Pickering et al. (2019)* |
| | *2.27–1.70 Ma (U-Pb) | *Edwards et al. (2023)* |
| **Bolt's Farm:** X Cave | <1.17 Ma (bovids) | *Van Zyl, Badenhorst & Brink (2016)* |
| **Bolt's Farm:** Pit 1, 3, 6, 25 | 2.33–0.78 Ma (fauna) | *Edwards et al. (2019)* |
| | 1.8 Ma (suids) | *Pickford & Gommery (2020)* |
| | 2.1–1.6 Ma (primates) | *Frost et al. (2022)* |
| **Bolt's Farm:** Pit 10 | <3.7 Ma (fauna) | *Edwards et al. (2019)* |
| | 1.9–0.7 Ma (primates) | *Frost et al. (2022)* |
| **Bolt's Farm:** Pit 16 | 2.33–0.99 Ma (equids) | *Edwards et al. (2019)* |
| **Bolt's Farm:** Pit 23 | 3.03–2.58 (fauna) | *Edwards et al. (2019)* |
| **Cooper's Cave** | 1.9–1.6 Ma (fauna) | *Steininger, Berger & Kuhn (2008)* |
| | 1.6–<1.4 Ma (fauna) | *De Ruiter et al. (2009)* |
| | *1.5–1.4 Ma (U-Pb) | *De Ruiter et al. (2009)* |
| | *1.62–1.4 Ma (U-Pb) | *Pickering et al. (2011)* |
| | *1.3 Ma (U-Pb) | *Pickering et al. (2019)* |
| | 1.9–1.6 Ma (primates) | *Frost et al. (2022)* |
| | 2.0–1.0 Ma (fauna) | *Hanon et al. (2022a), Hanon et al. (2022b)* |
| **Drimolen:** Main Quarry | 2.0–1.5 Ma (fauna) | *Keyser et al. (2000), Moggi-Cecchi et al. (2010)* |
| | 1.89–1.6 Ma (fauna) | *Adams et al. (2016)* |
| | 1.7 Ma (ESR) | *Herries et al. (2019), Herries et al. (2020)* |
| | *2.0–1.82 Ma (U-Pb) | *Pickering et al. (2019)* |
| | *2.04–1.95 Ma (Combo) | *Martin et al. (2021)* |
| **Drimolen:** Makondo | 2.7 Ma (ESR) | *Herries et al. (2019)* |
| | <2.6 Ma (fauna) | *Herries et al. (2018)* |
| | *2.62–2.28 Ma (U-Pb) | *Pickering et al. (2019)* |
| **Haasgat** | ~2.5 Ma (fauna) | *Keyser (1991)* |
| | 1.5–1 Ma (fauna) | *Plug & Keyser (1994)* |
| | 2.3–1.9 Ma (fauna) | *Adams (2012)* |

**Table 2** (*continued*)

| Site | Published dates | Authors |
|---|---|---|
| | 2.6–1.8 Ma (PM) | *Herries et al. (2014)* |
| | *2.0–1.82 Ma & 2.17–2.12 Ma (U-Pb) | *Pickering et al. (2019)* |
| | 2.1–1.7 Ma (primates) | *Frost et al. (2022)* |
| **Hoogland** | 3.12–2.58 Ma (fauna & PM) | *Adams et al. (2010)* |
| | *7.25–5.33 Ma (U-Pb & magnetostratigraphy) | *Hopley et al. (2019)* |
| | *3.08–2.83 Ma & 2.62–2.29 Ma (U-Pb) | *Pickering et al. (2019)* |
| **Malapa** | 2.36–1.5 Ma (fauna) | *Dirks et al. (2010)* |
| | *1.95–1.78 Ma (U-Pb) | *Dirks et al. (2010), Pickering et al. (2011)* |
| | ∼2.3–2.0 Ma (primates) | *Gilbert et al. (2015)* |
| | *2.0–1.82 Ma (U-Pb) | *Pickering et al. (2019)* |
| **Sterkfontein:** Jakovec Cavern | 3.5 Ma (fauna) | *Clarke (2002a)* |
| | <3.5 Ma (fauna) | *Berger, Lacruz & De Ruiter (2002)* |
| | ∼4.0–3.76 Ma (CN) | *Partridge et al. (2003)* |
| | <2.36 Ma (equid) | *Kibii (2004)* |
| | 3.61 Ma (CN) | *Granger et al. (2015)* |
| **Sterkfontein:** Member 2 | 3.5–3.0 Ma (carnivores) | *Turner (1997)* |
| | 3.3–3.33 Ma (PM) | *Partridge et al. (1999)* |
| | <3 Ma (fauna) | *Berger, Lacruz & De Ruiter (2002)* |
| | 4.17 Ma (CN) | *Partridge et al. (2003)* |
| | *2.2 Ma (U-Pb) | *Walker, Cliff & Latham (2006)* |
| | *2.8–2.6 Ma (U-Pb & U-Th) | *Pickering & Kramers (2010)* |
| | 2.6–1.8 Ma (ESR & PM) | *Herries & Shaw (2011)* |
| | *2.62–2.28 Ma (U-Pb) | *Pickering et al. (2019)* |
| **Sterkfontein:** Silberberg Grotto | *2.2 Ma (U-Pb) | *Walker, Cliff & Latham (2006) Pickering & Kramers (2010)* |
| | 2.6–2.2 Ma (PM) | *Herries & Shaw (2011)* |
| | 3.67 Ma (CN) | *Granger et al. (2015)* |
| | *2.3–2.1 Ma (U-Pb) | *Pickering et al. (2019)* |
| **Sterkfontein:** Member 4 | 2.8–2.4 Ma (bovids) | *Vrba (1975), Vrba (1980)* |
| | ∼2.5 Ma (primates) | *Delson (1984), Delson (1988)* |
| | 2.8–2.6 Ma (fauna) | *McKee (1993)* |
| | ∼2.1 Ma (ESR) | *Schwarcz, Grun & Tobias (1994)* |
| | 2.5–1.5 Ma (fauna) | *Berger, Lacruz & De Ruiter (2002)* |
| | 2.15–2.14 Ma (PM) | *Partridge (2005)* |
| | *2.65–2.01 Ma (U-Pb & U-Th) | *Pickering & Kramers (2010)* |
| | 2.8–2.0 Ma (ESR & PM) | *Herries & Shaw (2011)* |

**Table 2** (*continued*)

| Site | Published dates | Authors |
|---|---|---|
| | *2.62–2.28 Ma & 2.17–2.12 Ma (U-Pb) | *Pickering et al. (2019)* |
| | 3.4 Ma (CN) | *Granger et al. (2022)* |
| | *2.61–2.07 Ma (ESR & U-Pb) | *Pickering & Herries (2020)* |
| **Sterkfontein:** Member 5 East | 2.0–1.7 Ma (fauna & archaeology) | *Kuman & Clarke (2000)* |
| | *1.4–1.1 Ma (dating seriation) | *Herries, Curnoe & Adams (2009)* |
| | 1.4–1.3 Ma (ESR & PM) | *Herries & Shaw (2011)* |
| | 2.1 Ma (CN) | *Granger et al. (2015)* |
| | *2.0–1.82 Ma (U-Pb) | *Pickering et al. (2019)* |
| | 2.5–1.8 Ma (primates) | *Frost et al. (2022)* |
| **Sterkfontein:** Member 5 West | 1.7–1.4 Ma (fauna & archaeology) | *Kuman & Clarke (2000)* |
| | *1.3–0.8 Ma (dating seriation) | *Herries, Curnoe & Adams (2009)* |
| | 1.3–1.1 Ma (ESR & PM) | *Herries & Shaw (2011)* |
| | *2.0–1.82 Ma (U-Pb) | *Pickering et al. (2019)* |
| **Sterkfontein:** Post Member 6 Infill | ~250–100 ka (fauna & archaeology) | *Reynolds, Clarke & Kuman (2007)* |
| | ~470–290 ka (ESR & PM) | *Herries & Shaw (2011)* |
| **Sterkfontein:** Lincoln Cave North | *~253–115 ka (U-Th) | *Reynolds et al. (2003)* |
| **Sterkfontein:** Lincoln Cave South | ~250–100 ka (fauna & archaeology) | *Reynolds, Clarke & Kuman (2007)* |
| **Swartkrans:** Member 1 Lower Bank | 2–1 Ma (bovids) | *Vrba (1975)* |
| | 1.8–1.5 Ma (fauna) | *Vrba (1985)* |
| | <1.9 Ma (primates) | *Delson (1988)* |
| | 1.73 Ma (equids) | *Churcher & Watson (1993)* |
| | *1.83 Ma (U-Pb of teeth, 60% error range) | *Balter et al. (2008)* |
| | 2.36–1.65 Ma or 2.1–2.0 Ma (fauna) | *Herries, Curnoe & Adams (2009)* |
| | *~2.25 Ma (U-Pb) | *Pickering et al. (2011)* |
| | 2.2–1.8 Ma (CN) | *Gibbon et al. (2014)* |
| | 2.22 Ma (CN) | *Kuman et al. (2021)* |
| | 2.0–1.8 Ma (primates) | *Frost et al. (2022)* |
| **Swartkrans:** Member 1 Hanging Remnant | 2–1 Ma (bovids) | *Vrba (1975)* |
| | 1.8–1.5 Ma (fauna) | *Vrba (1985)* |
| | <1.9 Ma (primates) | *Delson (1988)* |
| | 1.73 Ma (equids) | *Churcher & Watson (1993)* |
| | ~2.11 Ma–1.6 Ma (ESR) | *Curnoe et al. (2001)* |
| | *1.83 Ma (U-Pb of teeth, 60% error range) | *Balter et al. (2008)* |

**Table 2** (*continued*)

| Site | Published dates | Authors |
|---|---|---|
|  | 2.36–1.65 Ma or 2.1–2.0 Ma (fauna) | *Herries, Curnoe & Adams (2009)* |
|  | *1.8–1.71 Ma (U-Pb) | *Pickering et al. (2011), Pickering et al. (2012)* |
|  | 2.0–1.8 Ma (primates) | *Frost et al. (2022)* |
| **Swartkrans: Member 2** | 1.5–1 Ma (fauna) | *Brain & Watson (1992)* |
|  | 1.1 Ma (fauna) | *Vrba (1995)* |
|  | ~1.6 Ma (fauna) | *De Ruiter (2003)* |
|  | *1.36 Ma (U-Pb of teeth) | *Balter et al. (2008)* |
|  | *1.63–1.41 Ma (U-Pb) | *Pickering et al. (2019)* |
|  | 2.7–1.2 Ma (primates) | *Frost et al. (2022)* |
| **Swartkrans: Member 3** | >1.65 Ma (primates) | *Delson (1988)* |
|  | 1.5–1 Ma (fauna) | *Brain & Watson (1992)* |
|  | 700–600 ka (fauna) | *Vrba (1995)* |
|  | <1.6 Ma (primates) | *De Ruiter (2003)* |
|  | *830 ka (U-Pb of teeth) | *Balter et al. (2008)* |
|  | 0.96 Ma (CN) | *Gibbon et al. (2014)* |
|  | 1.8–1.2 Ma (primates) | *Frost et al. (2022)* |
| **Swartkrans: Member 4** | *~110 ka (U-Th) | *Sutton et al. (2009)* |
| **Swartkrans: Member 5** | *<11 ka ($C^{14}$) | *Brain (1993)* |

well as primates (*Adams et al., 2010*; *Hopley et al., 2019*). Robert Broom first explored a cave system in the Schurveberg region in the mid-1930s, where he collected various fossil specimens from *ex situ* breccias, including the *Papio* (*Dinopithecus*) *ingens* type specimen (*Broom, 1936*) as well as the smaller *P. robinsoni* (*Freedman, 1957*). The site of Hoogland was later postulated to be the source for part of Broom's 'Schurveberg collection', though this is not certain, and discoveries at Sterkfontein, Swartkrans and Kromdraai meant Broom did not return to the site. Hoogland, an active cave system with exposed deposits, was eventually visited in 2008, when the first *in situ* excavations took place (*Adams et al., 2010*). This was also when the first official processing of the *ex situ* fossils from Broom's collection was conducted—over seventy years after the specimens were collected. The sole primate reported by *Adams et al. (2010)*, *Theropithecus oswaldi*, is not present in Broom's assemblage in the Ditsong Museum of Natural History. The current faunal sample (Table S6) is mostly derived from the *ex situ* deposits, and the bovids show significant taxonomic diversity. Other than the original report on initial fossil discoveries (*Adams et al., 2010*), no further faunal reporting has been undertaken for Hoogland.

### Biochronology and absolute dates

The faunal analyses coupled with palaeomagnetic results gave a provisional age of 3.12–2.58 Ma for the site (*Adams et al., 2010*). However, primate biochronology deemed this estimate to be unlikely if Schurveberg and Hoogland are the same deposit, with *D. ingens* and *P. robinsoni* limiting this range to 2.1–1.7 Ma (*Frost et al., 2022*). *Pickering et al. (2019)* U-Pb dated the basal speleothem to 3.145 ± 0.243 Ma, which is comparable to

the faunal and palaeomagnetic age of *Adams et al. (2010)*. However, *Hopley et al. (2019)* used magnetostratigraphy on the thick basal speleothem to give a Miocene age between 7.25–5.33 Ma, based on the trace element record of the same speleothem. They went on to use U-Pb laser ablation dating to a final corrected age of $5.28 \pm 0.12$ Ma, but acknowledged that their correction is based on regional $^{234}U/^{238}U$ measurements, rather than measuring this uranium isotope ration in their U-Pb dated samples, and as such, they acknowledged the possibility that their age actually spans a range of 1 Ma, sitting between 4.8 and 5.8 Ma. *Pickering et al. (2019)* were able to measure the residual $^{234}U/^{238}U$ disequilibrium in their sample of this same basal flowstone, and using this direct measurement (and not a regional average like *Hopley et al., 2019*), they were able to resolve the age to $3.145 \pm 0.243$ Ma. Given the thickness of this basal flowstone, several meters, and the presence of fine white laminations, there are likely many breaks in flowstone growth, and we have no firm grasp of the number nor length of these hiatuses, so it is not inconceivable that this flowstone formed intermittently from 5.0 to 3.0 Ma. Of the most importance here is that the faunal collection from Hoogland *overlies* this flowstone, and given the *Pickering et al. (2019)* date, it cannot be older than ~3.1 Ma, which is consistent with the existing biochronology for the deposit. More careful field sampling and additional U-Pb work in the future will resolve this apparent discrepancy.

## Malapa
### Faunal reporting
Malapa is one of the Cradle's more recently discovered fossil localities (*Berger et al., 2010*; *Dirks et al., 2010*; *Kuhn et al., 2011*), known for having yielded at least two partial skeletons attributed to *Australopitheus sediba* (*Berger et al., 2010*; *De Ruiter et al., 2013*; *Irish et al., 2013*). It is so far the only site where the species has been found. The main pit deposit consists of various facies (A through E) within an eroded surface infill (*Dirks et al., 2010*). Upon first exploration of the area, 209 non-hominin fossils were collected from facies D–E, though just 25 specimens were identifiable to genus level (*Dirks et al., 2010*). Later analyses of the entire vertebrate collection, including specimens from what are noted as 'unknown' facies, indicated 1,302 fossil specimens, of which 971 were identifiable to family level (*Val et al., 2015*). Other than *Papio angusticeps*, which has been described in detail (*Gilbert et al., 2015*), most of the Malapa faunal collection (Table S7) has either yet to be formally examined, or the focus has been on under-represented species such as suids and small carnivores (*Lazagabaster et al., 2018*; *Van der Merwe, Baker & Kuhn, 2021*). Bovids, the most abundant faunal group at Malapa, were analysed after excavations resumed in 2015, where it was noted that previous identifications were made without providing accession numbers (*Brophy et al., 2016*), thus direct comparisons of specimens was not possible. The species *Makapania broomi* is of particular interest in the analysis of Malapa bovids—fossils attributed to *Makapania* sp. are found in three Sterkfontein Members, Swartkrans Member 1, and Haasgat. It is thus unclear whether various species of *Makapania* were present during the Plio-Pleistocene at the Cradle, or whether these were in fact morphotypes of the same genus and species.

*Biochronology and absolute dates*

The initial recovery of extant representative carnivore taxa (*Felis silvetris*, *Parahyaena brunnea* and *Lycaon* sp.), a bovid (*Tragelaphus* cf. *strepsiceros*), and *Equus* sp. from Malapa placed the site's maximum age at around 2.36 Ma, and presence of the extinct carnivore *Megantereon whitei* has a LAD in Africa 1.5 Ma, providing a faunal age bracket for the deposits (*Dirks et al., 2010*; *Kuhn et al., 2011*). However, these FADs and LADs assigned to the abovementioned fauna were all derived from correlations with eastern African faunal assemblages and are thus not entirely reliable. Specifically, the first recorded appearance for *Equus* in Africa at 2.36 Ma was from eastern Africa (*Berger, Lacruz & De Ruiter, 2002*; *Herries & Shaw, 2011*), and this date in particular has been used routinely to provide upper and lower age limits at South African sites. The LAD for *Megantereon whitei* of 1.5 Ma is also problematic. In fact, the species was reported at the South African coastal site Elandsfontein, dated to between 1 Ma and 600 ka, also *via* comparisons to absolute dates from the east (*Klein et al., 2007*). The only non-hominin primate recovered from Malapa (*P. hamadryas angusticeps*) was noted years after the site's first discovery, and had biochronological implications with a FAD for the species at Haasgat argued to be ~2.4–2.0 Ma (*Gilbert et al., 2018*; *Gilbert et al., 2015*). The flowstones at Malapa suggest a narrow depositional time window for the fossils that aligns with the biochronology (Fig. 4), between 2.062 ± 0.021 Ma and 2.048 ± 0.140 Ma (*Dirks et al., 2010*; *Pickering et al., 2011*). The palaeomagnetic polarities of the sequence were used to narrow this time window down to 1.98 Ma (*Pickering et al., 2011*).

# Sterkfontein
## Faunal reporting

Sterkfontein is the best known of the Cradle sites and remains the richest source of australopithecine fossils in the world. Following the discovery of archaic stone tools, the site was first divided into the older Type Site (Sts), where artefacts were absent, and the younger Extension Site (ES) (*Robinson & Mason, 1957*). Later, Robinson recognised a third, or youngest, breccia on top and renamed them as the Lower, Middle and Upper breccias (*Robinson, 1962*). *Partridge (1978)* finally divided the total depth of breccia into six 'Members'—Member 1 to 3 in the lower caves, the Silberberg Grotto and the Jakovec cavern, and Member 4 to 6 in the surface exposures (*Clarke, 2006*). The initially proposed 'layer-cake' model for Sterkfontein, which suggested that deepest deposits were the oldest layers and the surface deposits the youngest, was later reconsidered, revealing the complex nature of the infill (*Clarke, 2006*; *Herries, Curnoe & Adams, 2009*; *Pickering & Kramers, 2010*).

One of the most iconic *A. africanus* fossils from Sterkfontein, the cranium Sts 5 colloquially known as Mrs. Ples, was described by *Broom (1947)* from the Type Site (Sts), which contains Member 4 (*Partridge, 1978*). Other notable hominins also come from Member 4, the small-bodied Sts 14 and the larger StW 431 that might represent different species (*Fornai et al., 2021*; *Grine, 2019*; *Kibii & Clarke, 2003*; *Macho et al., 2020*). Sterkfontein is also widely known for the remarkably complete 'Little Foot' skeleton StW 573, found in the Silberberg Grotto of Member 2 (*Clarke, 1998*; *Clarke, 2019*; *Clarke &*

*Kuman, 2019*; *Clarke & Tobias, 1995*). The younger Member 5 deposit at Sterkfontein has yielded various stone tools as well as the extensively studied StW 53 cranium, initially thought to belong to *Homo* (*Hughes & Tobias, 1977*; *Kimbel, Johanson & Rak, 1997*), though more recently it has been recognised that the StW 53 infill is a mix of Member 4 and Member 5 deposits and that the StW cranium belongs to a late representative of *A. africanus* not associated with stone tools (*Clarke, 1995*; *Clarke et al., 2021*; *Prat, 2004*; *Zanolli et al., 2022*).

Sterkfontein was noted early on as containing 111 bovid specimens within the Sts alone (*Vrba, 1974a*), 42 of which came from the ES to the West, excavated by Robinson in 1957/8 (*Robinson, 1962*). It was previously suggested that the ES may be of older depositional age than the Sts, with the overall difference in bovid composition within each deposit taken as confirmation of their progression in time (*Vrba, 1974a*; *Vrba, 1975*). The carnivores at Sterkfontein were described by *Turner (1987)* and examined again after further excavations (*O'Regan, 2007*; *O'Regan & Reynolds, 2009*; *Turner, 1997*). The reanalyses and inclusion of new material resulted in a large diversification of carnivore species lists from all levels of the site (*O'Regan & Reynolds, 2009*) (Table S8).

The deepest Sterkfontein deposit, the Jakovec Cavern (we use this spelling as first used by *Wilkinson (1973)*), produced various primate species (including australopithecines) as well as an equid (*Kibii, 2004*; *Partridge et al., 2003*). Jakovec Cavern includes five Carnivora families, and bovid tribes of various ecological niches, from grassland adapted to those occupying more closed habitats (Table 1). Notable tribes represented are Hippotragini, Reduncini, Cephalophini, Tragelaphini, Bovini and Alcelaphini. Excavation of Member 2 breccia revealed various *in situ* and *ex situ* fauna (*Delson, 1984*; *Delson, 1988*; *Pickering, Clarke & Heaton, 2004*; *Tobias, 1979*). Important specimens from this deposit include two hyaena species of the genus *Chasmaporthetes*, four extinct primate species (*Papio izodi*, *Parapapio jonesi*, *Parapapio broomi* and *Cercopithecoides williamsi*), and the extinct bovid *Makapania broomi*. Apart from *Turner (1997)*, earlier studies that purportedly described fauna from Member 2 also included fossils from lime miner dumps deposited during operations in the Silberberg Grotto—which also contains breccia from Member 3—thus caution should be exercised regarding these previous taxonomic lists for Member 2 (*Broom, 1939*; *Clarke, 1994*; *Pickering, Clarke & Heaton, 2004*; *Tobias, 1979*).

Members 4 and 5 have been described jointly in terms of fauna, and *McKee (1991)* provided a breakdown of the bovids, carnivores and cercopithecids from these deposits. Including the famous Sts 5 cranium, most *A. africanus* fossils, as well as three *Parapapio* species (*P. broomi*, *P. jonesi*, *P whitei*), *Dinopithecus ingens* and *C. williamsi* have been documented from Member 4 (*Clarke, 2013*; *Clarke & Kuman, 2019*; *De Ruiter, 2004*; *Freedman, 1957*; *Freedman & Stenhouse, 1972*; *Heaton, 2006*; *Kibii, 2004*; *Kuman & Clarke, 2000*; *Mokokwe, 2016*). Carnivores and bovids are also well represented in this Member (Fig. 2), including various extinct species. Noteworthy bovids recovered from Member 4 include *Makapania broomi*, *Megalotragus* sp., two extinct *Antidorcas* species (*A. recki* and *A.* cf. *bondi*), *Hippotragus cookei*, *Redunca darti*, *Pelea capreolus*, and *Syncerus* sp. (*Kibii, 2004*; *Reynolds & Kibii, 2011*). The extensive fauna from this Member have, in part,

been used for palaeoenvironmental reconstructions (*Elton, 2001*; *Luyt & Lee-Thorp, 2003*; *O'Regan & Reynolds, 2009*; *Reed, 1997*; *Vrba, 1974a*; *Vrba, 1975*; *Vrba, 1985*; *Vrba, 1995*).

Member 5 fauna as well as stratigraphy demonstrated that the deposit is composed of three separate units of different ages (*Kuman & Clarke, 2000*). However, research has demonstrated the possibility that the distinct Stw 53 infill is likely a remnant of Member 4 (*Couzens, 2021*; *Granger et al., 2022*; *Kuman & Clarke, 2000*; *Mokokwe, 2016*; *Pickering, 1999*). Consequently, Member 5 is discussed here only in terms of its East and West deposits. Member 5 East has yielded three extinct primate species, five species of carnivores, and bovids represented by Bovini, Antilopini, and Alcelaphini tribes (*O'Regan, 2007*; *Pickering, 1999*). Various hominin specimens have come from Member 5 West, as well as one extinct carnivore species, and Bovini, Tragelaphini, Antilopini, Alcelaphini and Aepycerotini bovid tribes (*Ogola, 2009*; *Pickering, 1999*). The faunal composition of Member 5 East is noted as overall more grassland-adapted than the fauna from Member 4 (*Avery, 2001*; *Kuman & Clarke, 2000*).

The Post Member 6 infill contains various fauna, with no extinct species demonstrated from the deposit (*Kuman & Clarke, 2000*; *Ogola, 2009*). Lincoln Cave, which also contains Post Member 6 infills, produced, other than hominin specimens attributed to *H. ergaster* (which indicate likely mixing between deposits) and a *Megalotragus* sp. specimen, no extinct fauna in either the North or South units (*Reynolds, Clarke & Kuman, 2007*; *Reynolds et al., 2003*).

Despite Sterkfontein containing perhaps the richest deposits of Plio-Pleistocene mammalian fauna in the Cradle region—with over eighty recognized species so far designated (*Reynolds & Kibii, 2011*), exhaustive faunal reporting has not been conducted, with many mammal groups of different members currently without published reports.

### Biochronology and absolute dates

While certain non-primate/non-carnivore faunal assemblages at Sterkfontein have perhaps not been given their due attention in recent years (Fig. 3 & Table S3), the dates of members and their stratigraphic relationships certainly have (Fig. 4). Dating *via* cosmogenic nuclides first provided a date of ∼4.0–3.76 Ma for Jakovec Cavern (*Partridge et al., 2003*), and later a date of 3.61 Ma (*Granger et al., 2022*).

The dating of Member 2 in the Silberberg Grotto has caused much debate over the years, due to the remarkably complete Little Foot skeleton (StW 573) recently attributed to *A. prometheus* (*Clarke & Kuman, 2019*; *Clarke & Tobias, 1995*). Initial biochronology placed the deposit at an estimated 3.5–3.0 Ma based on the presence of a *Chasmaporthetes* fossil, an extinct hyaena, with affinities to *C. australis* found at the Pliocene site of Langebaanweg (*Clarke, 1998*; *Turner, 1997*), while cosmogenic nuclide burial methods suggested an even older age of 4.17 Ma (*Partridge et al., 2003*). Conversely, *Berger, Lacruz & De Ruiter (2002)* even ascribed an age of 1.95–1.07 Ma to the fossils. This young date has since been countered (*Clarke, 2002a*; *Pickering & Kramers, 2010*; *Walker, Cliff & Latham, 2006*), while absolute U-Pb methods produced an age of ∼2.2 Ma for Member 2 (*Pickering & Kramers, 2010*; *Walker, Cliff & Latham, 2006*). *Granger et al. (2015)* then repeated cosmogenic nuclide dating within the deposit and proposed an age of 3.61 Ma,

whereas palaeomagnetic stratigraphy for this unit of Member 2 suggested a range of 2.6–2.2 Ma (*Herries & Shaw, 2011*), and the refined U-Pb regional chronology indicated a younger range of 2.17–2.14 (*Pickering et al., 2019*). *Clarke (2002b)* and later *Bruxelles et al. (2019)* argued that the flowstone associated with StW 573 in Silberberg is intrusive and thus postdates the skeleton. However, according to *Edwards et al. (2023)*, there is currently no concrete evidence that flowstones can be intrusive, as no evidence for "solutional unconformable contacts" (*Granger et al., 2022*: 4) has been provided, nor have photomicrographs. This makes this an unlikely explanation for the age discrepancies.

Faunal dating for Member 4 has relied on eastern African correlations, with an age of ∼2.5 Ma based on the cercopithecids (*Delson, 1984*; *Delson, 1988*), 2.8–2.4 Ma based on the bovids (*Vrba, 1975*; *Vrba, 1980*), and 2.5–1.5 Ma based on all fauna (*Berger, Lacruz & De Ruiter, 2002*). The presence of *Equus* suggested an upper limit of 2.36 Ma (*Herries & Shaw, 2011*). ESR and palaeomagnetic dating proposed that Member 4 is ∼2.1 Ma (*Schwarcz, Grun & Tobias, 1994*), or 2.15–2.14 Ma (*Partridge, 2005*), or 2.8–2.0 Ma (*Herries & Shaw, 2011*). U-Pb dating gave a range of 2.65–2.01 Ma for the deposit (*Pickering & Kramers, 2010*). A combination of published U-Pb, ESR and palaeomagnetic narrowed the age range of Member 4 down to 2.61–2.07 Ma (*Pickering & Herries, 2020*). Finally, based on cosmogenic nuclides, *Granger et al. (2022)* recently suggested a date of 3.4 Ma, which would be similar to Jakovec Cavern and the Silberberg Grotto. This age is at odds with various other proxies, particularly fauna, and has consequently been the subject of much debate (*Bibi, 2022*; *Frost et al., 2022*; *Frost et al., 2023*; *Granger et al., 2023*).

The use of biochronology for deposits such as these, where much contextual information has been lost and radiometric methods are at odds with other methods, can be particularly valuable. The faunal data most likely derive from similar sedimentary and taphonomic processes as the hominins (see *Clarke, 2019* for fauna directly associated with StW 573 in the Silberberg Grotto), and can thus be a reliable indicator for the history of sediment deposition. When radiometric ages have conflicted in the past (*e.g.*, *Curtis et al., 1975*; *Fitch et al., 1974*), faunal estimates have often proven to be crucial in determining which ages are more reliable (*Cooke, 1976*; *Harris & White, 1979*). For the Member 5 East sequence, biochronology and the presence of Oldowan stone tools placed the deposit at 2.0–1.7 (*Kuman & Clarke, 2000*). A multi-disciplinary seriation of dating techniques including ESR, palaeomagnetism and analysis of sedimentation time suggested that this Member 5 deposit is between 1.4–1.1 Ma (*Herries, Curnoe & Adams, 2009*), a range later refined to 1.4–1.2 Ma (*Herries & Shaw, 2011*). In contrast, the primate specimens alone indicated an age range of 2.5–1.8 Ma (*Frost et al., 2022*). This better agrees with a cosmogenic nuclide analysis that produced an age of 2.18 Ma (*Granger et al., 2015*), whereas *Pickering et al. (2019)* placed these deposits into the depositional window between 2.0–1.82 Ma *via* U-Pb flowstone analysis.

Member 5 West, also initially dated *via* biochronology and archaeology, *i.e.*, Acheulean stone tools, suggested a slightly younger age of 1.7–1.4 Ma (*Kuman & Clarke, 2000*). Further methods, both relative and absolute, proposed that the deposit is 1.26–0.82 Ma (*Herries, Curnoe & Adams, 2009*), or 1.3–1.1 Ma (*Herries & Shaw, 2011*); younger ranges than offered by faunal or archaeological data.

With regards to the Post Member 6 infill and the two Lincoln Cave units, these deposits have either been dated relatively or absolutely (*Herries & Shaw, 2011*; *Reynolds, Clarke & Kuman, 2007*; *Reynolds et al., 2003*), but none have been dated using both. Dates for the Post Member 6 Infill were determined *via* fauna, ESR and palaeomagnetism, and range between 470–100 ka (*Herries & Shaw, 2011*; *Reynolds, Clarke & Kuman, 2007*). The North Lincoln Cave was assigned an age of ∼253–115 ka based on uranium-thorium (U-Th) dating (*Reynolds et al., 2003*), and the South Cave was suggested to be slightly younger at ∼250–100 ka based on fauna and archaeology (*Reynolds, Clarke & Kuman, 2007*).

## Swartkrans
### *Faunal reporting*

As with Sterkfontein, Swartkrans was initially thought to be stratigraphically simpler than was later discovered (*Brain, 1981*; *Brain, 1993*), and was consequently divided into five distinct 'Members' following *Partridge (1978)*. Member 1, separated by *Brain (1993)* into two units—the Lower Bank and the Hanging Remnant—boasts the largest collection of *Paranthropus robustus* fossils ever recovered, with over 99 individuals derived from this member alone (*Brain, 1970*; *Brain et al., 1988*; *Broom & Robinson, 1949*; *Grine, 1993*; *Grine & Susman, 1991*; *Pickering et al., 2012*). Though less abundant, early *Homo* fossils are also found in this member (*Clarke, 1977*; *Clarke & Howell, 1972*; *Clarke, Howell & Brain, 1970*; *Pickering et al., 2012*; *Zanolli et al., 2022*). The co-occurrence of these fossils is among the earliest evidence of two early Pleistocene hominin species co-inhabiting the same region.

First explored in 1948 (*Broom & Robinson, 1949*), material from Swartkrans was published in the years that followed. The originally recovered fossils were regrouped in the late 1970s based on new evidence, which suggested they came from two distinctly aged breccias, subsequently named Member 1 and Member 2 (*Freedman & Brain, 1977*). The cercopithecids and bovids from the site were the only faunal specimens to be properly re-examined. The remaining fauna was not formally described until almost two decades later, despite excavations revealing a substantial assemblage that included 64 macromammalian taxa—later stated to come from four distinct members (*Watson, 1993*). The first evidence of a giraffid at the site was described by *Churcher (1974)*, though not attributed to any member. The same is true for early accounts of carnivores from Swartkrans (*Hendey, 1974*), simply described as being overall heterogeneous temporally. *Freedman (1957)* provided comprehensive descriptions of the primate material, and *Theopithecus oswaldi* and *Papio* cf. *robinsoni* are noted as present in all Swartkrans members (*Delson, 1984*; *Delson, 1988*; *Delson, 1993*). Vrba provided an extensive report on all bovid material from Member 1 (*Vrba, 1974b*) and Member 2 (*Vrba, 1974c*), delineating taxonomic classifications and detailed descriptions of features for each specimen. Amongst these, various tooth specimens cautiously marked as *Gazella* were described. Later, *Watson (1993)* reported on the faunal composition of Members 1–3, where many bovid horncore specimens also tentatively referred to as *Gazella* were recorded. However, the absence of dental or other material for *Gazella* sp. was noted and indeed contrasts with Vrba's earlier report. *Antidorcas*/*Gazella* isolated teeth can be difficult to distinguish, however, so this is not entirely surprising. A species of the extinct bovid genus *Megalotragus* was reported from Member 2, as well as a

diverse sample of bovid species that were of little use for biochronology. *Equus quagga* is notably present in Member 3, as well as the carnivore genus *Megantereon* that appeared to represent a relatively recent record for the genus (*Watson, 1993*). Hyracoidea were reported as appearing in all Members except Member 4, and the two initially reported species were later deemed conspecific (*McMahon & Thackeray, 1994*).

Many years after the initial recovery of fossils, and a decade after they were first suitably described, the overall faunal assemblage from Swartkrans (Table S9) was reanalysed (*De Ruiter, 2003*). Excavations recommenced in 1965 and the faunal sample increased over the next 21 years by around 350,000 specimens. Clarification of stratigraphy enabled the depositional placement of fossils previously deemed to be of 'uncertain origin' in the Swartkrans sequence (*Brain, 1993*). Improvements in identification since the original specimens were first described have also increased overall taxonomic level precision. While the analysis by *De Ruiter (2003)* is currently the most comprehensive record for Swartkrans, it is now two decades old.

### Biochronology and absolute dates

Prior to the division of Member 1 into two sub-deposits, faunal dating was assigned to the whole of Member 1 collectively and offered different dates depending on the study and the type of fauna analysed: 2–1 Ma based on bovids (*Vrba, 1975*), <1.9 Ma based on primates (*Delson, 1988*), 1.73 Ma based on an equid specimen (*Churcher & Watson, 1993*), and 1.8–1.5 Ma based on overall faunal composition (*Vrba, 1985*). Since the division of Member 1, separate analyses were possible. Two cosmogenic nuclide studies were conducted in the Lower Bank, both of which returned similar ages of ∼2.2 Ma (*Gibbon et al., 2014*; *Kuman et al., 2021*). An ESR study on one *Paranthropus* and two bovid teeth of the Hanging Remnant revealed a weighted mean age of 2.1–1.6 Ma (*Curnoe et al., 2001*). U-Pb was conducted on bovid teeth within both units, though with error ranges of over 60% ($1.82 \pm 1.38$ Ma) (*Albarède et al., 2006*; *Balter et al., 2008*). *Pickering et al. (2011)* dated the flowstone above and below both units and provided ages of 2.25 Ma for both basal flowstones and 1.8–1.71 for both capping flowstones, suggesting that the Hanging Remnant and Lower Bank were deposited during the same time window, between 2.25 and 1.7 Ma.

Regarding Members 2 and 3, the initial biochronological age was deemed 1.5–1 Ma by *Brain & Watson (1992)*, though *Vrba (1995)* considered there to be differences and consequently regarded Member 2 to be 1.1 Ma and Member 3 to be 700–600 ka. However, *De Ruiter (2003)* noted in his reanalysis of the overall assemblage that the fauna appeared contemporaneous in Members 1 through 3. *Gibbon et al. (2014)* dated Member 3 to $0.93 \pm 0.09$ Ma based on cosmogenic nuclides. U-Pb dating of fossil bovid teeth produced dates of $1.36 \pm 0.29$ Ma and $830 \pm 210$ ka for Members 2 and 3, respectively (*Balter et al., 2008*). *Pickering et al. (2019)* place Member 2 into the 1.63–1.41 Ma time window based on flowstone U-Pb dates.

## DISCUSSION

### Effects of incomplete Cradle faunal reports and changing approaches to cave geology

Original faunal reports for many of the Cradle's fossil sites are at worst outdated or at best they provide an incomplete description of assemblages. While there have been attempts to update the faunal reports of some sites in recent years, this is complicated by the loss/misplacement of specimens, as well as the loss of contextual information over the years. There are also incongruities in what data are given priority and why. For example, despite its clear biochronological relevance, fauna from Bolt's Farm's Waypoint 160 has not been reported on in detail, yet there have been various publications centred around the cave. Other pits at the site have received considerably less attention overall (*e.g.*, X Cave, Garage Ravine), though faunal reports or analyses have been conducted and published. Generally, carnivores and primates are given priority across sites (Fig. 3, Table S3).

Faunal reports that are decades old often contain species classifications that either no longer exist or require extensive updates due to new evidence. Ultimately, the composition of the assemblage may alter significantly, impacting how it is understood in terms of its palaeoenvironment or chronological interpretations. This has been evidenced with the Haasgat assemblage reanalysis (*Adams, 2012*), where the initially suggested forest environments were later discounted, as was the site's depositional age. While the primary faunal report for the overall site of Sterkfontein is exhaustive and inarguably thorough (*Kibii, 2004*), it is an unpublished thesis that is almost two decades old. Knowing what we do now about the reanalysis of fauna at other Cradle sites (*e.g.*, Cooper's Cave, Drimolen, Haasgat and Malapa), there are likely unrecorded specimens among this largest collection of macromammals for the region. Efforts should be taken to ensure that fossil mammal taxa are well studied and well described for both biochronological and palaeoenvironmental reasons, as focused specimen-based analyses of fauna have proven their value in advancing understanding at these important sites (*Bibi, 2022*).

Historically, stratigraphy that was generally broad and somewhat vague was provided for the Cradle's karst deposits and their associated faunal assemblages. The multidisciplinary application of U-Pb, palaeomagnetic and biochronological dating has revolutionised the chronology of caves in this region. Each site, including the ones omitted from this review (*e.g.*, Kromdraai, Gondolin and Gladysvale), has its own unique stratigraphic and depositional history (*Gommery et al., 2012*; *Makhubela & Mavuso, 2022*; *Rovinsky et al., 2015*; *Stratford, 2017*), but overall the understanding of the caves has advanced greatly in recent years (*Bruxelles, 2022*; *Pickering & Edwards, in press*). Stratigraphic complexity is still present, however, as has been evidenced in Sterkfontein's Milner Hall (*Stratford et al., 2012*; *Stratford, Grab & Pickering, 2014*) or in the localised spatial modification of StW 573 (*Clarke, 2006*; *Stratford, 2017*), but we stress that these are single chambers at one cave, *vs.* the strides in understanding the rest of the caves across the region. The deposits around where StW 431 were found are also complicated, as the area is decalcified and has undergone volume changes and localised mixing such that the fossil was found in two adjacent square yards but with a vertical distribution of 2.1 m (*Haeusler et al., 2020*).

A further complicating factor is that, in some instances, specimens originally recovered from one deposit have since been assigned to a different one (*e.g.*, *Clarke, 2006*; *Kuman & Clarke, 2000*; *Ogola, 2009* for StW 53), and the discrepancies between publications have confounded research when it comes to interpreting these assemblages and their palaeoenvironmental associations—evidenced by both the Haasgat faunal assemblage review (*Adams, 2012*; *Adams & Rovinsky, 2018*) and the reanalysis of Malapa fauna (*Brophy et al., 2016*). It is possible to remedy this issue by compiling historical accounts and offering universal delineations for specific provenies, such as the approach used by *Edwards et al. (2019)* at Bolt's Farm.

## Current understanding of Cradle palaeoenvironments from faunal records

Fossil fauna is a significant indicator of ancient climatic and environmental conditions at sites of human evolution. Accordingly, despite their arguably limited nature, faunal reports from the Cradle have been used widely for palaeoenvironmental reconstructions (*e.g.*, *Keyser, 1991*; *Kuhn et al., 2016*; *Kuman & Clarke, 2000*; *Pickering, Clarke & Heaton, 2004*). However, there are fewer of these studies than one might anticipate given the amount of fauna recovered at the Cradle, especially when compared to the number of similar studies from eastern African sites of comparable age. In Kenya's Koobi Fora Formation alone, many palaeoenvironmental studies have been conducted using fauna, and differences in palaeoecological conditions through time are estimated *via* proliferation and disappearance of various taxa (*Bobe & Carvalho, 2019*; *Bobe & Eck, 2001*; *Pobiner et al., 2008*; *Reed, 1997*). Of the eight key Cradle sites discussed here, palaeoenvironmental reconstructions for six of them (Bolt's Farm, Cooper's Cave, Drimolen, Haasgat, Hoogland and Malapa) are based mainly on faunal presence data (*Adams & Rovinsky, 2018*; *Badenhorst & Steininger, 2019*; *Gommery et al., 2008*; *Kuhn et al., 2011*; *Pickering, Clarke & Heaton, 2004*; *Watson, 1993*). Apart from Sterkfontein and Swartkrans, methodologies that directly extract environmental information from fauna (*e.g.*, carbon and oxygen isotopes, ecomorphology, dental wear) are scarcely conducted. When these types of studies are conducted, great insight is gained as to palaeoenvironmental conditions at Plio-Pleistocene sites (*Lee-Thorp, Sponheimer & Luyt, 2007*; *Luyt & Lee-Thorp, 2003*; *Peterson et al., 2018*; *Sewell et al., 2019*; *Steininger, 2011*), and they are worth carrying out on a wider scale.

### *Bolt's farm*

Other than one ecomorphology study for which no sample provenies were given (*Elton, 2001*), environmental conditions of Bolt's Farm pits have been implied based only on faunal assemblages. Waypoint 160 was suggested to be an open habitat environment based on primate postcrania (*Gommery et al., 2008*), and based on the presence of *Metridiochoerus*, *Phacochoerus*, and *Hippopotamus*; Milo A has been reconstructed as a woodland savanna with $C_4$ grasses and nearby water (*Gommery et al., 2012*). Based on hypsodont suids from Aves Cave and other nearby pits, this area was reconstructed as largely grassland (*Pickford & Gommery, 2016*; *Pickford & Gommery, 2020*); as were Garage Ravine (*Badenhorst et al., 2011*) and X Cave (*Van Zyl, Badenhorst & Brink, 2016*) based on the sample size of bovids and equids.

### Cooper's Cave

The majority of ungulates at Cooper's Cave (Alcelaphini, Antilopini, equids and *Metridiochoerus*) suggest a grassland environment, with other taxa (Bovini, Tragelaphini and Giraffidae) pointing to the presence of wooded and water components (*Badenhorst & Steininger, 2019*; *De Ruiter et al., 2009*; *Hanon et al., 2022b*). This varied range of vegetation, including wooded grasslands, woodland/bushland areas (see *Cerling et al., 2011* for terminology of vegetation structures) and water-adjacent habitats, has also been supported by the diverse and abundant carnivore assemblage at the site (*Kuhn, Werdelin & Steininger, 2017*; *O'Regan & Steininger, 2017*). Carbon isotope data from Cooper's Cave bovids suggested a mosaic environment too, with a significant proportion of mixed feeders as well as obligate $C_4$ and obligate $C_3$ feeders (*Steininger, 2011*).

### Drimolen

The Drimolen Main Quarry has a diverse, yet somewhat uninformative, faunal sample (*Adams et al., 2016*). Antilopini (*Antidorcas recki*—28.6% of the total assemblage) and Alcelaphini (*Connochaetes* sp., *Damaliscus* sp. and *Megalotragus* sp.) dominate the bovid assemblage, and are often associated with wooded grasslands (*Vrba, 1985*). However, the overall habitat suggested by faunal presence is broad, with most identifiable taxa occupying mixed and open-to-closed environments. DMQ is also notably rich in carnivores (mostly Felidae), totalling 12.9% of the overall assemblage (*Adams et al., 2016*). Similar to DMQ, Antilopini and Alcelaphini bovids also dominate DMK (*Rovinsky et al., 2015*), which like DMQ is regarded as a challenging environment to reconstruct (*Murszewski, Boschian & Herries, 2020*). This is partly due to a lack of identifiable craniodental specimens compared to many postcranial remains (*Rovinsky et al., 2015*), an issue that could be addressed using ecomorphological methods (*e.g.*, *Barr, 2014*; *Kappelman, 1988*).

### Haasgat

*Pelea capreolus* is the most abundant ruminant at Haasgat, signifying a wooded grassland environment. *Cercopithecoides williamsi* was noted as the most widely recovered animal overall, which *Keyser (1991)* interpreted as indicative of a more forested habitat. At other sites where *Cercopitheoides* occurs, however, it is often associated with open habitats (*Frost & Delson, 2002*). Later faunal analyses (*McKee & Keyser, 1994*; *Plug & Keyser, 1994*) suggested a mixed habitat occupied by grazers (Alcelaphini), browsers (Tragelaphini and Giraffidae) and wetland-typical taxa (*Kobus leche* and *Kobus ellipsiprymnus*). More recent palaeoenvironmental reconstructions argued against this interpretation, as *Kobus* is variable in its utilization of habitat and does not necessarily dictate the presence of wetlands (*Adams, 2012*; *Adams & Rovinsky, 2018*). Moreover, carbon isotope ratios from primates and non-primates indicated an extensive $C_3$ presence, as well as significant open-to-wooded grassland habitats (*Adams, Kegley & Krigbaum, 2013*).

### Hoogland

The palaeoenvironment of Hoogland is unresolved, though it has been tentatively suggested that the presence of both grazers (Alcelaphini and Reduncini) and browsers (Tragelaphini

and *Procavia transvaalensis*) within the deposits indicate both $C_3$ and $C_4$ vegetation was present during deposition (*Adams et al., 2010*).

### Malapa

The diverse and dominant carnivore assemblage from Malapa has indicated different habitat types. Extant *Felis* and Canidae are associated with grassland environments, as well as *Dinofelis* and *Atilax* which are associated with closed, wet environments (*Kuhn et al., 2016*; *Kuhn et al., 2011*). One suid specimen identified as *Metridiochoerus* (*Lazagabaster et al., 2018*) suggested a more open habitat (*Bishop, 2010*). Bovids, equids and primates are less abundant and thus not utilised in palaeoenvironmental reconstructions (*Kuhn et al., 2016*). However, support for a more forested/wooded environment comes from recently recovered Viverridae specimens, comprised of two species (*Genetta genetta* and *Genetta tigrina*) known to avoid grasslands and open habitats (*Van der Merwe, Baker & Kuhn, 2021*). Despite the evidence from carnivores pointing to wooded environments, bovid specimens from Malapa are more consistent with that of other Cradle sites; a mosaic of both grassland and woodland (*Brophy et al., 2016*). However, the bovid sample is notably smaller than the carnivore sample, so these reconstructions are less reliable.

### Sterkfontein

Faunal data from Sterkfontein was widely employed for early palaeoenvironmental reconstructions for the site (*e.g.*, *Kuman & Clarke, 2000*; *Vrba, 1975*; *Vrba, 1980*; *Vrba, 1985*). The dominant presence of Alcelaphini and Antilopini in Members 4 and 5 was assumed to indicate a mainly open plains and wooded grassland environment (*Vrba, 1975*), though the presence of *Makapania broomi* in Member 4 pointed to a more wooded component. *Reed (1997)* studied the Member 4 faunal community and suggested an open woodland based on the significant number of frugivorous animals. *Kuman & Clarke (2000)* demonstrated a change from more forest/woodland conditions in Member 4, to open woodland and grassland conditions in Member 5. Member 2, based on the mammalian assemblage, was interpreted as rock-littered and scrub-filled with nearby grassland (*Pickering, Clarke & Heaton, 2004*).

### Swartkrans

Bovid data presented by *Vrba (1975)* suggested an overall more open environment for Swartkrans than Sterkfontein Member 4, with the site said to represent the onset of a move towards more open environments in South Africa. *Watson (1993)* considered the overall faunal composition—and hence the environment—at Swartkrans to be unchanged across members, with species indicating a wooded grassland with rocky hills and riverine woodland savanna. *De Ruiter (2003)* noted that reconstructing the Swartkrans palaeoenvironment based on fauna is complex, as one cannot be certain that the assemblage represents a truly random sample of the community. Conversely, the environment could be interpreted as mosaic; the presence of Antilopini and Alcelaphini could imply a wooded grassland, elephants and other bovids inhabit woodland, and otters, water mongoose, and reedbuck are all present and suggest a nearby water source. As mentioned above, however, it is important to approach reconstructions of this kind with caution (*Sokolowski et al., 2023*).

Beyond faunal presence data, stable carbon isotopes is the most widely used tool in the Cradle for palaeoenvironmental reconstruction. Measurement of carbon isotopes (presented as $\delta^{13}C$ values) from mammalian tooth enamel enables the tracking of environmental changes at ancient sites by assigning the difference in isotopic signature to differences in vegetation outside the cave, specifically trees ($C_3$) *vs.* grasses ($C_4$) (*Lee-Thorp, Van der Merwe & Brain, 1989*; *Lee-Thorp, 2008*). One of the first carbon isotope studies conducted at the Cradle was on Swartkrans Member 1 baboons (*Lee-Thorp, Van der Merwe & Brain, 1989*) where it was found that one species (*Papio robinsoni*) had a $C_3$-based diet and the other (*Theropithecus darti*—now recognised as *T. oswaldi* (*Getahun, Delson & Seyoum, 2023*)) a $C_4$-based diet. In contrast, *Codron et al. (2005)* measured $\delta^{13}C$ ratios in extinct baboons from both Swartkrans and Sterkfontein, where they determined extensive use of $C_4$ grasses in all five taxa studied at both sites. Another early carbon isotope study is that of *Van der Merwe & Thackeray (1997)* at Sterkfontein, where carbon isotope ratios of bovids showed presence of both $C_4$ and $C_3$ vegetation. A later study based only on bovids at Sterkfontein indicated a shift to open environments through time at the site (*Luyt & Lee-Thorp, 2003*). Perhaps the best-known carbon isotope study of Cradle fauna is that of *Lee-Thorp, Sponheimer & Luyt (2007)*, who tracked environmental changes at Sterkfontein, Swartkrans and Makapansgat and concluded that $C_4$ grazing became more abundant after 1.7 Ma (*Lee-Thorp, Sponheimer & Luyt, 2007*). However, the dates within the study rely on eastern African faunal comparisons. Carbon ratios of bovids from Swartkrans Members 1–3 were measured by *Steininger (2011)*, who found that Member 1 represented a mosaic or $C_4$ wooded grassland, and that Members 2 and 3 were more heterogeneous with less $C_4$. This compares with two isotopic studies at Makapansgat Member 3 that also revealed more browsers than faunal abundance analyses had previously predicted (*Sponheimer & Lee-Thorp, 2003*; *Sponheimer, Reed & Lee-Thorp, 1999*). More recent carbon isotope analyses on small mammal fossils from Sterkfontein, Swartkrans and Gladysvale indicated a strong contribution of $C_4$ grasses overall (*Leichliter et al., 2017*), though with a higher $C_3$ signal for Swartkrans than *Lee-Thorp, Sponheimer & Luyt (2007)* determined. *Sewell et al. (2019)* recently used *Antidorcas* carbon isotope values as well as other dietary methods at Sterkfontein and Swartkrans, and suggested increased grass presence after 1.7 Ma, with Swartkrans Member 2 being more heterogenous. However, *Antidorcas* species are known to alternate between grasses in the wet season and dicotyledonous plants in the dry season (*Venter & Kalule-Sabiti, 2016*). Thus, this signal may indicate the persistence of palatable grasses throughout the seasonal cycle, as opposed to reflecting overall grass coverage of the landscape.

Ecomorphology is another technique that, while applied to some Cradle sites, has not been explored as fully and is a helpful technique for sites such as Drimolen which lack craniodental specimens. The method defines a relationship between a phenotypic characteristic, such as bone morphology, and habitat in extant taxa (*Barr, 2014*; *Kappelman et al., 1997*; *Plummer et al., 2015*), then applies that relationship to fossil specimens in order to interpret habitat preferences in extinct taxa (*Bishop et al., 2011*; *Forrest, Plummer & Raaum, 2018*). Since its inception, ecomorphology has been used extensively for palaeoenvironmental reconstructions in eastern Africa (*Barr, 2014*; *Faith et al., 2012*;

*Plummer, Bishop & Hertel, 2008*). Ecomorphology of cercopithecoid postcrania from Sterkfontein Member 4, Swartkrans Members 1 and 2 and Bolt's Farm, suggested a wide range of habitats at Sterkfontein and Bolt's Farm, including forest, mixed and open (*Elton, 2001*). However, despite some pits at Bolt's Farm being up to 2 million years apart, no provenience is given for any of the samples. Swartkrans specimens differed from the other sites in that no adaptations to forested environments were observed, suggesting that these cercopithecoids were mostly open-habitat dwellers (*Elton, 2001*), a finding which contrasts some of the isotopic evidence (*Lee-Thorp, Sponheimer & Luyt, 2007*; *Sewell et al., 2019*; *Steininger, 2011*).

Mesowear analysis of Plio-Pleistocene Cradle bovids (*Schubert et al., 2006*; *Sewell et al., 2019*) has not been conducted on a large scale, despite the speed and affordability offered by the method. Moreover, none of the mesowear studies conducted at the Cradle have taken a regional approach, and are instead conducted on a site-by-site basis. *Antidorcas* mesowear and microwear from Sterkfontein and Swartkrans indicated an increase in grassland presence from 1.7 Ma, in accordance with carbon isotope results from the same study (*Sewell et al., 2019*) and those of *Lee-Thorp, Sponheimer & Luyt (2007)*. Again, however, this signal could only be reflecting the persistence of palatable grasses across seasons.

The apparent scarcity of such direct analyses at the Cradle is also likely due in large part to the taphonomic influences in the caves, where the effects of time- and climate-averaging impacts the feasibility of such studies. Faunal assemblages at the Cradle likely accumulated under varying climatic regimes (*Behrensmeyer, Kidwell & Gastaldo, 2000*; *Hopley et al., 2013*; *Reed, 1997*), resulting in an artificially large biodiversity and effecting palaeoecological interpretations (*Hopley & Maslin, 2010*). Similarly, it is accepted that faunal assemblages are also time-averaged on scales up to tens of thousands of years (*Kidwell & Behrensmeyer, 1993*; *Kingston, 2007*), thus assemblages can represent temporally unrelated communities (*Bobe, Alemseged & Behrensmeyer, 2007*; *Lyman, 2017*), or the effects of short term fluctuations in habitat can be masked (*Avery, 2007*; *Hopley & Maslin, 2010*).

Though not the focus of this paper, evidence from microfauna and birds point to similar conclusions as the macrofauna: a gradual movement towards a more grassland dominated environment, though without an incontrovertible date for when this shift began to take place (*Avery, 2001*; *Denys, 1992*; *Senegas, 2004*). The two rodent genera, *Mystromys* and *Otomys*, that dominate the micromammal assemblage at Cooper's Cave are grassland adapted (*Linchamps et al., 2023*), and the bird assemblage is noted as indicating a savanna environment with rocky outcrops and woodland (*Pavia et al., 2022*). Elephant shrew evidence at Malapa also pointed to a savanna or shrubland environment (*Val et al., 2011*). Various micromammal species from Sterkfontein Member 5 and Swartkrans Members 1–3 suggested that the deposits were between wooded grassland and savanna biomes, and that Sterkfontein Member 4 was a riverine landscape with gallery forest (*Avery, 2001*). *Bamford (1999)* explored fossil wood from this Member, where presence of a liana species also suggested a forested palaeoenvironment. Vegetation at these two sites is suggested to have shifted from riverine grassland with bushes, to open savanna plains. Rodent taxa that are restricted to arid and semi-arid areas (*Petromus* and *Euryotomys*) are present at Bolt's Farm (*Senegas, 2004*; *Senegas & Avery, 1998*), which suggests a drier environment than today.

Overall, we have a regional picture of Cradle palaeoenvironments during punctuated records of fossil accumulation, which demonstrates repeated instances of wetter to drier environments based on the U-Pb chronology (*Pickering et al., 2019*). Within this there is mixed evidence of open grassland to closed and wooded environments, likely indicating a mosaic environment but with a progression towards grasslands. Faunal presence data are undoubtedly useful in palaeoenvironmental reconstructions, but widespread use of the aforementioned techniques would help us better understand conditions at the Cradle. Additionally, these investigations can now be performed on data with constrained dates that are separate from eastern African faunal correlations.

## Inferences from Cradle biochronology—was South Africa a refugium?

Biochronological inferences from fauna are helpful, though the fauna used needs an excellent reference dataset free from gaps, and this does not yet exist for South or eastern Africa. Regardless, there has been debate over the years as to whether various regions of Africa served as biotidal habitats (defined as areas of persistence and comparatively little change (*Gamble, 2009*)) or refugial habitats during the Plio-Pleistocene (*Vrba, 1988*). A biotidal record will demonstrate significant disappearances and first appearances of taxa during times of great climatic shift, whereas a refugial area is characterised by the persistence of its dominant taxa. It has been suggested that southern Africa acted as a refugium throughout Plio-Pleistocene climatic shifts, where species went extinct later than in eastern Africa. This was hypothesized because the southern region of the continent was thought to be less affected by the tectonic, volcanic and palaeolake processes that occurred in eastern Africa around this time (*Bailey, King & Manighetti, 2000*; *Partridge, Wood & De Menocal, 1995*). *Reynolds (2007a)* aimed to test this by examining body size changes (proxies for climatic changes) in modern and fossil Hyaenidae, Equidae, and Bovidae from both eastern and southern sites. It was argued that the southern conspecifics had larger body sizes, indicating improved fasting endurance, in turn making them more resilient to shifts in ecological structure than the eastern taxa (*Reynolds, 2007a*). Other species-specific studies also indicated that because climate change was less severe in the south than it was in the east, taxa persisted there when they did not at eastern sites (*Arctander, Johansen & Coutellec-Vreto, 1999*; *Brink & Lee-Thorp, 1992*; *Hay, 1990*). Based on genetic evidence, eastern populations of *Connochaetes taurinus* appear younger than the southern populations, suggesting recolonization after local extinction (*Arctander, Johansen & Coutellec-Vreto, 1999*). However, population genetics of the other Alcelaphini indicated the opposite trend, with the oldest populations coming from eastern Africa (*Arctander, Johansen & Coutellec-Vreto, 1999*). Another genetic study compared the population histories of eastern and southern *Taurotragus oryx*, and the southern lineage was found to have persisted for longer based on an earlier time to the most recent common ancestor (*Lorenzen et al., 2010*). DNA polymorphism data from *Equus quagga* also identified southern Africa as the region from which all populations of the plains zebra expanded from during the mid-Pleistocene (*Pederson et al., 2018*).

Biochronological dating of South African sites based on eastern African faunal correlations presents challenges if South Africa was in fact a refugium where species lasted

for longer than they did at eastern sites. A fitting example of this comes from Cooper's Cave, first dated *via* biochronology to be no younger than 1.6 Ma (*Berger et al., 2003*). This date was based on the presence of *Metridiochoerus andrewsi*, a species that last appears in Koobi Fora above the Okote Member tuff, dated to 1.56 Ma (*McDougall & Brown, 2008*). Later U-Pb dates, however, indicated a younger deposit (~1.4 Ma) for Cooper's Cave (Fig. 4), suggesting the species survived in South Africa for longer. Similarly, Swartkrans Member 3 contains *Chasmaporthetes*, where both its presence and that of *Megantereon* was noted as interesting as these taxa were not, at the time, known to persist in Africa so late in time (*Brain et al., 1988*; *Watson, 1993*).

While biochronological records are invaluable, narrower direct ages are ideal, with biochronological data used to support these. Age brackets for South African sites that are based on first and last appearance dates in eastern Africa are broad (*e.g.*, *Dirks et al., 2010*). Even if South Africa did not serve as a refugium during Plio-Pleistocene climatic change, these dates can be misleading in a different way, as they can also estimate younger depositional ages than is actually the case at South African sites. Again, however, the effects of time- and climate-averaging limit or can bias these interpretations somewhat (*Domínguez-Rodrigo & Musiba, 2010*; *Hopley et al., 2013*; *Hopley & Maslin, 2010*).

Though at times limited, biochronology remains a key element of dating Cradle sites, providing a test of direct methods (Table 2). For example, Sterkfontein Members 2, 4 and the Silberberg Grotto have all been suggested to be over 3 million years old based on cosmogenic nuclides (*Granger et al., 2015*; *Granger et al., 2022*), though no cercopithecids of this age or older (*i.e.*, *Parapapio ado*, *Cercopithecoides meaveae*, or smaller subspecies of the *Theropithecus oswaldi* lineage) are found at any Cradle sites (*Frost et al., 2022*). The oldest possible geological age estimate based on cercopithecoids from the Cradle is 3.0–2.0 Ma at Sterkfontein Member 2 (*Frost et al., 2022*), though there has also been debate surrounding this interpretation (*Frost et al., 2023*; *Granger et al., 2023*).

## How geochronology has improved our understanding of Cradle site ages

Providing precise geochronological ages for South Africa's Cradle sites was historically challenging, particularly the first attempts many decades ago (*Blackwell, 1994*; *Brain, 1993*; *Partridge & Watt, 1991*; *Vrba, 1975*). This difficulty was caused in part by the disruptive anthropogenic activity (primarily lime mining) that occurred at numerous Cradle sites (*Edwards et al., 2019*), and in part by the inability to employ the radiometric techniques commonly used to date eastern African sites (namely argon-argon and potassium-argon dating) due to an absence of volcanic activity. The lack of dateable ash layers in South Africa resulted in the belief that these sites were simply not directly dateable. Thus, for decades there were no attempts or investments to provide precise ages for the Cradle deposits beyond the existing biochronology, palaeomagnetism and early ESR dates (*Curnoe et al., 2001*; *Delson, 1988*; *Herries et al., 2006*; *Reynolds, Clarke & Kuman, 2007*; *Schwarcz, Grun & Tobias, 1994*; *Turner, 1997*).

Now that direct ages are attainable at the Cradle, evaluating biochronological dates in comparison to U-Pb dates is crucial for a comprehensive understanding of depositional

histories. For obvious reasons, a great deal more faunal dates than absolute ones have been reported for Cradle sites (Fig. 4, Table 2). Assigned ages often, and ideally, overlap across different methodologies, however discrepancies between methodologies have also been observed. U-Pb dating of Drimolen Main Quarry supplied an age of 2.0–1.8 Ma, a range which corresponds well with the faunal estimate (2.0–1.5 Ma) and helps to constrain potential periods of deposition. On the other hand, Sterkfontein Member 4 was previously dated to 2.8–2.6 Ma using fauna (*McKee, 1993*) whereas later direct methods placed the Member just outside of this range at 2.6–2.07 Ma (*Pickering & Herries, 2020*). Overall, however, when all absolute ranges and all faunal ranges are analysed in terms of inter-method agreement per site or member, results are mostly concordant. Sterkfontein Member 4 methods overlap in age ranges 85% of the time, with the discrepancy mostly due to the faunal range being broader in terms of the younger limit (Fig. 4). Sterkfontein's Silberberg Grotto estimates are also in accordance, though again with a broader faunal age (a million-year range *vs.* 0.2 Ma for absolute methods). For both Swartkrans Member 1 deposits, Member 2, and Malapa, there is strong inter-method agreement, though the absolute age ranges are much narrower. Interestingly, they fall almost exactly in the middle of the overall faunal range. At DMK the methods are in near-complete agreement, with the absolute methods providing a wider range since the faunal estimate is represented by a single age. The overlap for Cooper's Cave sits at just under 50%, a discrepancy that is largely attributable to the fauna estimating an age 0.3 Ma older than the absolute methods for the older limit, and 0.1 Ma older for the younger limit. Agreement at Haasgat is just under 40%, in part due to the very narrow absolute age (2.17–1.82 Ma) and some faunal estimates being much younger overall (1.5–1.0 Ma). This younger (1.0 Ma) estimate for the bovids may warrant further attention, as it is based mainly on the absence of extinct bovid species that typically occur in Plio-Pleistocene deposits. No agreement is observed in Hoogland estimates, where the oldest faunal age is 0.1 Ma younger than the youngest absolute age. The limited faunal composition at Hoogland likely contributes to the inconsistency in dating, and future work at the site may enhance its biochronological value.

U-Pb dating is now not only the most widely employed absolute dating method across all Cradle sites, but indeed appears to be the most precise option to provide direct age brackets for fossil deposition. The method is particularly reliable when it is applied as part of a multidisciplinary system in which various relative methods are also utilised.

## CONCLUSION

Faunal reports from the cave sites of South Africa's Cradle of Humankind have historically been underexplored. Additionally, assemblages, particularly those of bovid taxa, are often left unanalysed despite their value in reconstructing landscapes. Primates and carnivores, on the other hand, are the focus of more work after initial taxonomic identification. When faunal reports that include bovid taxa are revised, these mammals routinely appear to be the ones with the highest number of unrecorded specimens. Moreover, there are instances of specimens initially recorded that, upon revision, are no longer locatable.

Both faunal dating and direct methods have been integral in forming the chronology for the Cradle area, though since fauna were previously utilised for biochronology they were

not evaluated in their own right. Due to the advent and introduction of locally appropriate dating, this is no longer the case and fauna can be explored independently and directly. While biochronologically derived ages for sites are often in line with absolute methods, they remain broad, a limitation that can be refined with improved faunal reporting and more extensive comparative analyses.

Most Cradle sites' palaeoenvironmental reconstructions, based primarily on faunal abundance data, have demonstrated a shift towards a grassland-dominated landscape from a more wooded one during the Plio-Pleistocene transition, though with evidence for surrounding water and woodland areas. Some sites also demonstrate more heterogeneity than others in their palaeoecological structure, such as Cooper's Cave. These conclusions are supported by further lines of evidence, including carbon isotopes and ecomorphology.

There is a clear need for the (non-primate) fauna, specifically the numerically-dominant bovids, at the Plio-Pleistocene hominin sites in South Africa's Cradle to be empirically examined when they are first recovered, as well as reviewed more regularly. Doing so will not only limit the often-incongruous species reports, but it will be invaluable in future attempts to reconstruct the palaeoenvironments of this region.

## ACKNOWLEDGEMENTS

We would like to thank the Ditsong National Museum of Natural History and the Evolutionary Studies Institute at the University of the Witwatersrand for access to their collections. We thank three anonymous reviewers for their critical but constructive comments which helped clarify and improve this manuscript. This research contributes towards the output of the Biogeochemistry Research Infrastructure Platform (BIOGRIP), established by the Department of Science and Innovation, South Africa.

### Funding

This work was supported by the Swiss Federal Commission for Scholarships, the Swiss Society for Anthropology, and the University of Zürich Graduate Research Campus. The funders had no role in study design, data collection and analysis, decision to publish, or preparation of the manuscript.

### Grant Disclosures

The following grant information was disclosed by the authors:
The Swiss Federal Commission for Scholarships.
Swiss Society for Anthropology.
University of Zürich Graduate Research Campus.

### Competing Interests

The authors declare there are no competing interests.

## Author Contributions

- Megan Malherbe conceived and designed the experiments, performed the experiments, analyzed the data, prepared figures and/or tables, authored or reviewed drafts of the article, and approved the final draft.
- Robyn Pickering analyzed the data, authored or reviewed drafts of the article, and approved the final draft.
- Deano Stynder analyzed the data, authored or reviewed drafts of the article, and approved the final draft.
- Martin Haeusler analyzed the data, authored or reviewed drafts of the article, and approved the final draft.

## Data Availability

This is a literature review.

## Supplemental Information

Supplemental information for this article can be found online at http://dx.doi.org/10.7717/peerj.18946#supplemental-information.

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
