# Peer review of "The large mammal fossil fauna of the Cradle of Humankind, South Africa: a review"

_PeerJ, doi:10.7717/peerj.18946_

## Round 0.1 · original submission · Major Revisions

Dear Dr. Malherbe,

I have now received the three reviews of your paper. As you can see, all the reviewers raised some important aspects to consider when revising the paper.

Regards,
Shaw Badenhorst

Reviewer 1 ·

Basic reporting

see below

Experimental design

see below

Validity of the findings

see below

Additional comments

Thank you for the opportunity to review this manuscript. The authors provide a historical overview of the large mammal faunas from the Cradle of Humankind sites, examining previous attempts to estimate the ages of the fossil deposits via biochronology and exploring the faunal-based paleoenvironmental interpretations. As noted by the authors, research on the large mammals from the Cradle is widely scattered and there are important gaps in research attention that demand further work. Because of this, as well the long history of uncertainty over the ages of these sites, the faunal records from the Cradle have received less attention than their eastern African counterparts, where access to robust age estimates and syntheses of the faunal data make it relatively easy to explore long-term evolutionary and environmental trends. This manuscript is a welcome and long over-due contribution, with potential to reinvigorate faunal research in the Cradle. With a bit more work, I think this has potential to be a very useful and widely cited paper. As you’ll see below, I raise a few substantive comments that need attention, all in the spirit (I hope) of trying to make this an even better reference for the research community.

Site Histories: The overviews on site histories focus on faunal reporting and biochronology. For all of the sites discussed here, it would be helpful to begin with a brief history of work at the site and a brief overview of the site and stratigraphy. Without these details, we are assuming a level of familiarity with the records that the reader may not have. For example, I can’t claim to be particularly familiar with the records from Bolt’s Farm, so when the text mentions Pit 4, Pit 5, Waypoint 160, etc. it doesn’t resonate. Similarly, there is little contextual information about Haasgat. In some cases, these details are woven through the other discussions, but I think a standalone overview would increase the overall utility of this paper. Again, brief is fine.

Site chronologies: The discussions of efforts to establish ages for the Cradle sites illustrates one of the reasons that have limited research attention on their larger mammal faunas. The ages are all over the place! Not much to do about that as it’s part of the research history, but you rightfully point out later on in the manuscript that U-Pb ages are revolutionizing our understanding of Cradle chronology. In the second half of the manuscript, where you pull together a synthesis of the research, it would be great to see further discussion of how the biochronological efforts relate to the absolute ages. Are they generally in agreement? Are there key sources of disagreement that merit further attention? The discussion beginning on Line 940 sets this up, and I was somewhat disappointed not to see a more detailed evaluation of how the faunal dates relate to the U-Pb dates. Your Figure 4 provides a great starting point.

Faunal data: It would be great to include a supplemental table providing updated taxonomic lists for each site / member. If that were done, I would refer to it and cite this paper all the time. It would be an incredible resource and it would greatly boost of the utility of this manuscript. It appears to me that you’ve already done the necessary research to put the pieces together. Taking this one step further would be well worth the effort.


Minor edits:
General comment on environmental terms: I think it’s important to be precise about terminology when describing paleoenvironments. For instance, you use the term grassland a lot, yet if we follow some widely used definitions (e.g., UNESCO woody cover categories: grassland = < 10% woody cover; for an application see (Cerling et al., 2011), true grassland is probably quite rare. Moreover, the associations between certain indicator taxa (e.g., Alcelaphini) and ‘grasslands’ are less clear than is widely assumed (Sokolowski et al., 2023)
Line 22: change “as true” to “true”
Line 34: probably replace “forested” with “wooded” – the latter is the term used later on in the manuscript and is probably more accurate.
Line 50: “countless” is hyperbole. Replace with something else.
Line 57-60: what about biochronology? That probably deserves mention here.
Line 66: “…incomplete or contradictory” Can you provide a few refs?
Line 70: I’d swap “assignations” to “attributions”
Line 71: I’d swap “can change” to “can be revised”
Line 77: Does not “aridity levels” fall under the umbrella of “climatic conditions”? To me it does, and you could probably just cut it.
Line 93-107: The focus here is on dietary reconstruction. What about paleoenvironmental work focused on community composition? Work by Vrba, Reed, and others comes to mind.
Line 157-160: Did you not also search the site names?
Line 214: Change “A. bondi and A. helmoedi” to “An. Bondi” and “Ae. helmoedi” This is convention when abbreviating genus names that start with the same first letter.
Line 232: “one of the oldest paleontological sites in South Africa.” That’s not true! See, for example, the fossil-bearing deposits from the Karoo Supergroup or most of the paleontology talks at the PSSA meetings.
Line 293: what was it about the faunal composition that led to this age estimate?
Line 337: “alcelaphine” to “alcelaphin” Informal reference to tribe names should end in ‘n’ (e.g., hominin for tribe Hominini), whereas ‘ne’ is used for subfamily (e.g., hominine for subfamily Homininae).
Line 353: what was it about faunal composition that suggested an age no older than 1.5 Ma?
Line 358: include a few references after “…archaeological deposits in southern Africa”
Line 396: Care to comment on the speleothem age estimate? That seems completely incompatible with the faunas.
Line 442: “australopithecine” to “australopith”
Line 482: you can delete reference to “Makapania sp.” since you already have “Makapania broomi) – the former doesn’t add taxonomic information.
Line 506: “acutely” is the wrong word for this
Line 595: “burchelli” is now subsumed within “quagga”
Line 693: can you provide refs for the paleoenvironmental research alluded to here?
Line 713: “support” to “suggest”
Line 715: what is “wide-ranging vegetative structure”?
Line 724: replace semi-colon with comma
Line 725-726: trim the ‘e’ at the end of Antilipin and Alcelaphin
Line 736-738: I’d replace “swamp” with “wetland”
Line 754: Why does Metridiochoerus signify a closed environment? I was under the impression that most species of this genus, and M. andrewsi in particular, are grazers (Bishop, 2010)
Line 811-814 and 833-836: Here the diets of springbok are interpreted in terms of overall grassiness on the landscape. Since springbok typically switch between grass in the wet season and dicots in the dry, might this instead be informing on the persistence of palatable grasses (rather than amount of grass on the landscape) through the seasonal cycle?
Line 877: I know Vrba has used it, but biotidal is not a widely used term. Google Scholar gives me only 306 results. Please define.
Line 891-893: This is confusing. Results imply one thing for blue wildebeest, but another for the Alcelaphini, a group that also includes blue wildebeest. Do you mean “other Alcelaphini”?


Bishop, L.C., 2010. Suoidea, in: Werdelin, L., Sanders, W.J. (Eds.), Cenozoic Mammals of Africa. University of California Press, Berkeley, pp. 821-842.
Cerling, T.E., Wynn, J.G., Andanje, S.A., Bird, M.I., Korir, D.M., Levin, N.E., Mace, W., Macharia, A.N., Quade, J., Remien, C.H., 2011. Woody cover and hominin environments in the past 6 million years. Nature 476, 51-56.
Sokolowski, K., Codding, B., Du, A., Faith, J., 2023. Do grazers equal grasslands? Strengthening paleoenvironmental inferences through analysis of present-day African mammals. Palaeogeography Palaeoclimatology Palaeoecology 629.

Reviewer 2 ·

Basic reporting

Here is my review of the manuscript entitled ‘The large mammal fossil fauna of the Cradle of Humankind, South Africa: a review’ submitted by Malherbe et al for consideration for publication in PeerJ.
According to the title of the paper, I was firstly delighted to see this manuscript which aims to produce a synthesis of the Plio-Pleistocene faunas of the Cradle of Humankind. This type of work is all too rare, even though syntheses and updates are essential, particularly concerning faunal material, which has not been treated with the same care as palaeoanthropological remains, while also taking into account the difficulties in chronologically aligning the deposits.
However, as I read the manuscript, I realised that this document does not provide any real novelty or vision, other than a partial state of the art. Please see my detailed comments below.
This article summarises the faunas of eight deposits from the Cradle namely Bolt's Farm, Cooper's, Drimolen, Haasgat, Hoogland, Malapa, Sterkfontein and Swartkrans. However, I am surprised to note in the abstract that the authors focus only on the sites (or members) for which dating (Uranium-Lead) has been produced. In the conclusion, the authors state that the biochronological contribution is in line with absolute methods. In my opinion, this is circular reasoning, given that the authors only limit themselves to sites with Uranium-Lead dates. Why not have tested the analysis on these sites and then tested the model with sites using other dating methods or even undated sites (but with a biochronological framework) in order to validate the approach? But we will come back to these conclusions later.
In their introduction, the authors state that ‘A faunal report, as discussed in this review, details the faunal remains found at a site and is frequently, though not always, focused on a single taxon or group’. This observation is partly true, but not completely. In recent years, there have been an increasing number of complete faunal lists in the literature that could have been incorporated into the present study (I am thinking here of Kromdraai, which has been the subject of several general and specific publications providing a complete and up-to-date faunal list, or the recent book ‘African Paleoecology’ edited by Reynolds and Bobe which includes number of up-to-date data). The authors go on to say that ‘faunal reports are often later found to be incomplete or contradictory, highlighting a lack of cohesive reporting at times. This, by extension, highlights the potential for still-unrecognised incongruities in faunal assemblages and reports that have gone without analysis or re-examination since their first mentions’, it's true that faunal assemblages need to be re-examined and reassessed, especially older works. However, this re-examination could also affect the most recent works, as palaeontological identification is largely based on the researcher's experience. In this sense, this argument is applicable to any site, any period and any geographical area. For example, to mention the sites presented in this article, in what way is Malapa's palaeontological work more consistent than Kromdraai's? The last sentence of the introduction must be rewritten and precise ‘Faunal reporting can also yield inferences about early hominin behaviour, as features like cut marks and the type of bone accumulation can demonstrate subsistence methods’. As such, it's not faunal reporting that makes it possible to identify cut marks or accumulation methods, but specific taphonomic or archaeozoological studies, which are not the most common.
In section 1.1, the authors focused the first part on bovid to infer their importance in palaeoenvironmental definition. However, as the authors point out later on, South African bovids are also found in East African contexts, raising the question of important geographical areas for these taxa or, on the contrary, refuge zones in southern Africa, thus limiting the biochronological and palaeoenvironmental implications of such a south-east africa comparison. Why not mention the case of smaller species, not to mention microfauna, such as herpestids, viverrids and mustelids, which, although not abundant in the assemblages, are extremely informative in terms of the palaeoenvironmental framework because they indicate palaeoecosystemic contexts on a regional scale?
From line 126, the authors mention the use of sites dated by Uranium-Lead as a solution for dating palaeontological fillings. I am still surprised that the authors do not really take other methods into account, or suggest that U-Pb dating appears as the main answer of the problem of dating the Cradle sites, particularly in view of the many discussions and debates about the consistence of dating. I am not discussing the date itself, but the question of contextual data, the dated element and the stratigraphic framework (I refer you here to the publications, debates and sometimes contradictory results of the work by Bruxelles, Herries and Pickering).
About section 2, I am very surprised by the comments of the authors considering the publication of Pickering et al 2019 as the absolute answer justifying the use and the presentation in this article of the only sites dated by Pickering. Scientific awareness and the veracity of scientific debates should have encouraged the authors to mention the response to this publication published by Stratford et al. 2020 or Granger et al. 2023. It is all too easy to sidestep scientific exchanges and disagreements by forgetting to mention publications and replies. Finally, the authors state that Large mammal fauna is the core focus. I'm sorry, but that in no way justifies excluding other taxa such as herpestids. The authors must produce consistent justifications for each of their methodological choices.
Line 231, the authors explain that the Aves complex at Bolt's Farm could be one of the oldest palaeontological sites in South Africa according to palaeomagnetic and U-Pb data with an age of between 3.03 and 2.61 Ma... what about Silberberg Grotto at Sterkfontein?
Lines 254-258, if the authors want to provide a summary of the detailed palaeontological research at Cooper's D, they should cite O'Regan et al 2013 and Cohen et al 2019 on mustelids, viverrids and herpestids.
The case of Hoogland biochronology and absolute dates is a perfect example. The authors show a certain consistency between palaeomagnetic dates and U-Pb. However, they also refer to Hopley's work, which places the site in the Miocene. In this particular case, which highlights the difficulty of dating South African sites, what date do the authors choose and, more importantly, why?
The authors repeatedly mention species of the genus Chasmaporthetes (including related specimens from sites indicating the species Chasmaporthetes silberbergi). However, when presenting the DMQ faunal species they use the taxonomic nomenclature used by Adams et al. 2016 Lycaenops silberbergi which is synonymous with C. silberbergi. I know this article is not focused on carnivore palaeontology, but the authors should homogenise the taxonomic nomenclature to make it clearer for the reader.
With regard to Chasmaporthetes from Member 2, neither Turner nor Clarke assigns the specimens to Chasmaporthetes cf. australis. Turner, while clearly assigning skull S94-13225 to C. nitidula, states that the specimen shares characters with Langebaan's C. australis, a plesiomorphic character suggesting potential phylogenetic relationships.
About discussion and debates on datings of Member 2, please summarize the arguments provided by Edwards et al 2023. This is all the more important as Edwards' publication only refers to the publication of Granger et al 2015 and 2022 and barely refers to the description of the stratigraphic sequence produced by Bruxelles et al 2019.
I'm surprised to find, following section 3 on the historical background and research into the study sites, a section 4 on discussion with no real presentation of the results. The results and the discussion should be clearly separated. In addition, the analysis methods are not clearly presented in a separate section. In fact, on reading section 4, I realise that this manuscript does not really present any original results, but is more of a bibliographical summary of the sites (which is partial because it focuses only on eight localities). In the conclusion, when the authors begin this section with the sentence ‘Faunal reports from the cave sites of South Africa’s Cradle of Humankind have historically been underexplored’, I regret to say that the present publication does not address this problem any further.
In my opinion, while this publication has the merit of reviewing the state of the art in terms of dating methods (although the manuscript concentrates almost exclusively on U-Pb dating) and faunal assemblages, it is too partial (because it is restricted to 8 sites with U-Pb dating) and effectively excludes other localities, even though (bio)chronological dates do exist.
This work does not offer anything new, nor does it provide a well-constructed analysis aimed at demonstrating a particular aspect.
As it stands, I cannot accept this manuscript for publication because it is simply a state of the art report with no other significant contribution.

Experimental design

see report

Validity of the findings

see report

Additional comments

see report

Reviewer 3 ·

Basic reporting

This is an important review article that highlights the need for more timely and consistent reporting of large mammal fossils (and ideally other forms as well) from the Plio-Pleistocene karst deposits of the Cradle. I can not support this contention strongly enough. The authors raise a key issue in paleoanthropology.

Of the review criteria given for this section, the one that most importantly needs addressing here is more thorough and systematic inclusion of key references. These are covered more specifically in section 4. of the review below. I've no doubt missed some, as I've commented mostly on primate references as I'm less familiar with the other taxa, but they will require similar more careful inclusion of references.

I feel the other review criteria are met by this submission.

Experimental design

This review is well-laid out, organized and clearly written. My main concern here is about bias (see comments in section 4 for specifics). In general, it reads as biased against biochronological methods as the weaknesses are often emphasized without also pointing out its utility and strengths. As mentioned in part 1 above, another area that needs addressing is more careful and thorough review of relevant literature.

Validity of the findings

This paper comes to a well-supported conclusion, and definitely identifies an unresolved issue that desperately needs to be addressed, and I thank the authors for that!

Additional comments

I'm putting minor and/or more specific comments here. Line numbers refer to those in the review pdf.

Introduction - Should include biochronology in the reasons faunal reports are important. This is a major theme of the article and feels strangely lacking in this section.

Lines 103-105 - Should really mention that this form of ecomorphology typically focuses on postcrania, and most often long bones, especially humeri and femora of various taxa, bovid astragali, etc. which are often challengin to identify below family level (e.g. bovid, suid, etc.) and are even more rarely commented upon in faunal reports that typically focus on which species are present and more taxonomically diagnostic elements.

Lines 93-107 - This paragraph covers diet and functional morphology well, but ignores faunal abundance studies. These should be discussed here, as they are frequently referred to later in the article.

Line 109 - Two references are given for dating using faunal comparisons. There are many more that should be included here (and are in fact used later in the manuscript). Please flesh out this list.

Lines 110-111 - This list of references seems really incomplete for a review article of this nature. Only covers the Turkana Basin, but ignores important sites like Olduvai, Tugen Hills, and all of the Afar region. Please update.

Line 115 - Biochronology definitely has drawbacks, but it is also worth emphasizing some its strengths. For example, unlike speleothems, while the fauna may be mixed or time-averaged, they are nonetheless in direct association with hominins and part of the same overall infill and likely accumulated due to similar processes. Also, it often requires far fewer models/assumption that are often problematic to measure or estimate (think ESR or cosmogenic nuclides). This is important, as one of the review criteria for review articles of this nature is to maintain an unbiased perspective.

Lines 120-122 - There is little evidence to really back the contention of South Africa as a refugium. The whole focus of this article is about how faunal reporting, and indeed dating of some key sites, is inadequate to reasonable evaluate hypotheses like this. While it is definitely possible, it is effectively conjecture at this point.

Lines 182-184 - Senegas & Avery, 1998 do not identify Papio at Waypoint 160 (or any primates for that matter). They base their age estimate on rodents. Gommery et al. (2008) identify Parapapio as present based on highly fragmentary set of specimens. It would be contradictory to use Papio as a justification for an early date as it has a first appearance ca. 2 - 2.4 Ma (see e.g. Delson, 1984; McKee, 1995; Heaton, 2006; Gilbert et al., 2018). Please rewrite.

Line 186 - Freedman, 1957 and Freedman, 1965 is the primary description of the cercopithecids from Bolt's. These references should be cited here.

Lines 204-205 - In addition to Pit 10, Frost et al., 2022 specifically give separate age estimates for Pit 6 and for Pit 23, as does Delson, 1984. These should be covered here as well.

Lines 218-221 - Cercopithecoides coronatus from Pit 6 would suggest a minimum age of approximately 1.5 Ma. See Delson, 1984; Frost et al., 2022.

Lines 224-226 - The presence of Papio would be contradictory here. FAD for Papio is ~2.4 Ma (see Delson, 1984; Gilbert et al., 2018; Frost et al., 2022). Further, Gommery et al. (2008) attribute these specimens to be Parapapio, not Papio. The specimens illustrated by Gommery et al, however, are not diagnostic below family.

Lines 230-232 - This statement is inaccurate. While the Aves complex may be among the oldest karst depostis in the cradle, it is half the age of Langebaanweg, and only a small fraction the age of sites in the Karoo.

Section 3.2.1 Cooper's Cave Faunal Reporting - This review section is not very complete. Primates for Cooper's "A", were thoroughly described by Freedman (1957). Material from newer excavations at Cooper's D have been described by Folinsbee and Reisz (2013) and DeSilva et al. (2013). Gilbert et al. (2018) reviewed Papio material from Cooper's A. This section needs revision.

Lines 261-262 - Delson (1984) also estimated the age of Cooper's "A". Should be included here.

Lines 301-303 - Papio robinsoni has a range of ~2.3 - 1.6 Ma. C. williamsi is ~2.7 - 1.6 Ma (see Frost et al., 2022). To say they are of no biochronological constraint is an overstatement. For example, we can say, that because DMQ has Papio robinsoni (and lacks Parapapio) it is younger than Makapan Mbs. 2-4 and Sterkfontein Mb 2 & 4. You make this exact point in the next sentence... Reword this sentence.

Section 3.5 Hoogland - It is important to point out that Broom reported the site as Skurweberg. Adams et al. (2010) are hypothesizing their site of Hoogland is the same, but we don't know that for certain. Furthermore, the sole primate reported by Adams et al., Theropithecus oswaldi is not found in Broom's assemblage in the Ditsong museum.

Lines 379-381 - Freedman also described Papio robinsoni from the site.

Lines 391-392 - Frost et al. 2022 found the age estimate of Adams et al. (2010) to be unlikely if Hoogland and Skurweberg are the same deposit. They suggested an age of 2.1 - 1.7 Ma. Please update.

Line 408 - Gilbert et al. (2015) report the presence (and thoroughly describe) of Papio angusticeps. Should mention this here.

Line 434 - Use Gilbert et al. (2018) for a more refined version of the age range for P. angusticeps (or P. h. angusticeps).

Section 3.7.1 Sterkfontein faunal reporting - It is inconsistent that the other fauna are discussed here in some detail, but primates are ignored. Freedman (1957), Freedman and Stenhouse (1972), Eisenhart (1974), and Heaton (2006) extensively described the primates from Sterkfontein Mb 4. Pickering et al (2004) those from Member 2. Not to mention reviews by Delson (1984; 1988), McKee (1995), Gilbert (2013), and Frost et al (2022). This matters as some of these reviews are discussed in the faunal dating section below, so their species content should be noted here along with the rest of the fauna.

Lines 532-543 - This section would be a logical place to re-iterate the point that biochronology can be particularly useful for collections that begin in the early to mid 20th century where much of the context has been lost, especially the position of fauna previously discovered (or from dumps) relative to speleothems: that is, at least the other fauna most probably derive from the same sediments and processes as the hominins because they are typically found in a co-mingled state (see Clarke, 2019 for Mb 2).

Lines 573-574 - I believe the earliest co-occurance demonstrated would be Zinjanthropus boisei and Homo habilis from Olduvai Bed I, including even at a single locality, F.L.K. I (Leakey et al., 1964).

Lines 575-598 - Freedman (1957) extensively described the primate material from Swartkrans. Also reviewed in Delson, 1984; 1988. Some Theropithecus material also described by Delson, 1993. These descriptions should be reviewed in this section as well.

Line 724 - Abundance data would make a big difference here as well as more accurate IDs.

Line 729 - Some references here for the type of ecomorphological studies referred to would help the reader.

Lines 732-734 - Keyser (1991) is misguided here. At other sites where it is known, Cercopithecoides is most often associated with more open habitats (e.g. Middle Bed II Olduvai, KBS member, Makaamitalu, etc.).

Lines 791-794 - This is a great example of taxonomy needing to be updated. Current scholarship universally recognizes Theropithecus from Swartkrans as T. oswaldi (or T. oswaldi oswaldi) and restricts T. darti (or T. o. darti) to Makapan and its eastern contemporaries - See Getahun et al., 2023 and references therein for a review.

Lines 900-902 - As discussed above, however, any evidence for South Africa as a refugium is based on faunal descriptions, which as made clear above, are far from adequate for this task. I think you make an excellent case, that on current evidence, there is no support for or against South Africa being a refugium.

Lines 923-926 - Frost et al., 2022 date Swartkrans Mb 1 to 2.1-1.7 Ma, and Makapan as the oldest Plio-Pleistocene site in South Africa at 2.7-2.6 Ma. For the Cradle, they estimate a max range for Sterkfontein Mb. 2 as 3.0 - 2.0 Ma, but suggest it is probably close in age to Member 4 which they suggest should be 2.6-2.0.

Lines 965-970 - It is true biochronological dates are often broad (though could be improved with better faunal reporting and broader comparative analyses), but they are rarely unreliable. In fact, they are often the clearest sign that a problem exists with other methods or stratigraphic/depositional interpretations.

---

## Round 0.2 · Minor Revisions

The authors addressed the comments of the reviewers, with just a few minor errors to correct.

Reviewer 1 ·

Basic reporting

see below

Experimental design

see below

Validity of the findings

see below

Additional comments

I appreciate the effort that went into the revision. I have only a few minor comments to add:

I see that Reviewer 2 took issue with the focus on sites associated with U-Pb chronologies, rather than all of the sites within the Cradle. Given the attention paid to biochronology in the text, I think that you could further justify this by explicitly pointing out that this subset of sites allows you to evaluate the faunal-based age estimates in light of absolute ages—i.e., the decision to look at sites with U-Pb chronologies is not arbitrary but rather allows you to address questions that are otherwise intractable. A small addition somewhere between lines 147-186 would do the trick.

Line 85: “methods of accumulation” is an odd way of phrasing it. Do you mean hunting versus scavenging? Or agents of bone accumulation?

Line 567: Sorry for not catching this earlier. On your list of noteworthy bovids from Member 4, I see you have “Damaliscus parmularis.” That’s noteworthy because there is no such species! I don’t have access to Kibii 2004, but I see that Reynolds and Kibii (2011) list “Damaliscus parmularius” in their Table 5 and then in text refer to it by a different spelling as “Damaliscus parmularis”. Parmularius is an extinct alcelaphin genus that is sister to Damaliscus, but to my knowledge there is no Damaliscus parmularus/parmularius. Most likely this was an accidental mistake. If not, and I've missed something somewhere in the literature, it's certainly not an accepted species (so i'd not mention it)

Line 842: Cercopithecoides needs to be italicized

Line 927: “grass volume” – volume implies 3 dimensions. How about “grass extent”?

---

## Round 0.3 · accepted · Accept

The authors addressed the minor changes the reviewer requested.

I also invited the reviewer who rejected the first version of the paper to review the revised paper, but the reviewer declined the invitation. This particular reviewer's main concern was that the paper was not making a significant contribution. However, considering that no such review exists of the large mammal remains in the Cradle of Humankind, I believe the paper is making a major contribution. The Cradle of Humankind in South Africa is one of the most important areas in the world for the study of hominin evolution, and this first review of the large mammals is a crucial step for the wider understanding of our species. This view, that the paper is making a significant scientific contribution that warrants publication, is also supported by the two other reviewers. In addition, the authors addressed all the other (relatively minor) concerns raised by the particular reviewer.